# CoLLMLight: Cooperative Large Language Model Agents for Network-Wide Traffic Signal Control

**Zirui Yuan**   **Siqi Lai**   **Hao Liu**[*]
Artificial Intelligence Thrust, The Hong Kong University of Science and Technology (Guangzhou)
{zyuan779, slai125}@connect.hkust-gz.edu.cn, liuh@ust.hk

## Abstract

Large Language Models (LLMs) have recently emerged as promising agents for Traffic Signal Control (TSC) due to their strengths in reasoning and generalization. However, current LLM-based approaches treat intersections as independent agents without inter-intersection cooperation, limiting their effectiveness in network-wide optimization. To address this gap, we propose CoLLMLight, the first cooperative LLM agent framework for network-wide traffic signal control. CoLLMLight enables agents to perform in-depth spatiotemporal reasoning for cooperation, while ensuring real-time responsiveness through an asynchronous cooperative decision architecture. The reasoning process runs asynchronously, deriving cooperative control guidance from dynamic interactions among intersections. This guidance is cached and incorporated as contextual input for real-time signal decisions. To enhance cooperation quality while ensuring reasoning efficiency, we propose cost-aware cooperation optimization. It first applies adaptive reasoning chain optimization to enable the LLM to adjust its reasoning depth according to traffic complexity. The model is then refined with reinforcement learning using reward signals that promote network-wide performance while penalizing excessive reasoning. Extensive experiments on four real-world traffic networks demonstrate that CoLLMLight consistently outperforms existing methods, achieving more effective and generalizable cooperation while maintaining real-time responsiveness and efficient token usage.

## 1 Introduction

Traffic congestion has become a pressing urban challenge, with profound societal and environmental impacts. As cities continue to grow due to rapid urbanization, the burden on transportation infrastructure intensifies. In this context, Traffic Signal Control (TSC) plays a pivotal role in improving traffic efficiency and road safety (Wu et al., 2023; Zhang et al., 2024; Wei et al., 2019c).

Over the past decades, both transportation-based and learning-based approaches have been explored to enhance TSC systems. However, these methods often struggle to generalize across diverse traffic environments. Recently, Large Language Models (LLMs) have attracted increasing attention in TSC and broader urban applications, leading to a series of LLM-based agents (Lai et al., 2025; Wang et al., 2024; 2025; Feng et al., 2024; Lai et al., 2026; Zhou et al., 2026; Han et al., 2025). Compared with prior TSC methods, LLM-based methods leverage language-driven reasoning, which improves interpretability and facilitates generalization across varying traffic conditions.

Despite their advancement, a fundamental limitation of these LLM-based agents is that they lack communication and act independently at a single intersection. This design ignores the cooperation across intersections and leads to a higher risk of network congestion. An example of this limitation is shown in Figure 1 (a): the independent agent selects the East-West phase to clear the longest local queue. While this decision appears reasonable from a local perspective, it ultimately causes upstream spillback, severely undermining overall traffic efficiency. In contrast, the cooperative agent (Figure 1 (b)) can leverage information from adjacent intersections to anticipate oncoming traffic,

---

[*]Corresponding author

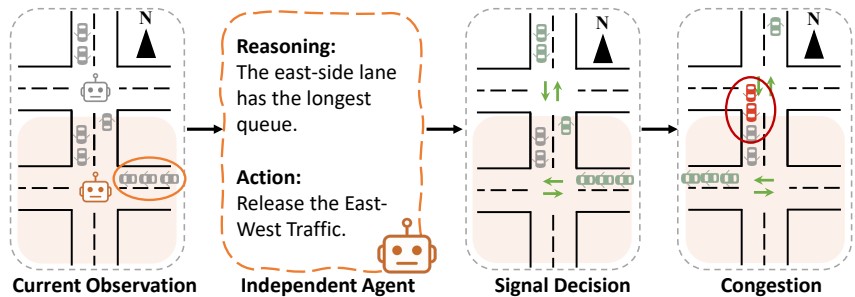

(a) Independent agent. This agent clears the longest queue based solely on local observation, leading to upstream blockage.

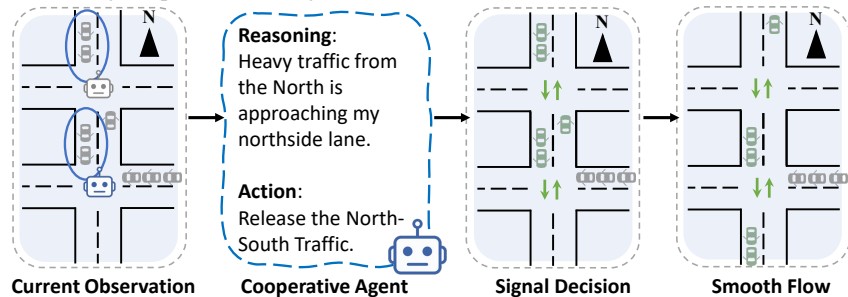

(b) Cooperative agent. This agent considers upstream traffic to proactively avoid congestion and ensure smooth flow.

Figure 1: **Independent vs. cooperative TSC agents**. Vehicles are colored by state: queued (gray), moving (green), and blocked (red). Green arrows indicate the selected signal; shaded areas denote each agent's observation range.

and proactively clears its lane to coordinate with upstream flows, thereby avoiding congestion and maintaining smoother network traffic.

While cooperative LLM-based TSC agents hold great promise, they also face several challenges. First, cooperation is inherently more complex than single-intersection control. Agents must reason about dynamic interactions among intersections and anticipate how local actions influence upstream and downstream flows, making it difficult to identify effective cooperation signals. Second, current LLMs perform multi-step reasoning for complex problems, which brings significant computational overhead and latency. This conflicts with the strict real-time requirements of traffic signal control, where decisions must be made within a few seconds. The third challenge lies in balancing cooperation quality with reasoning efficiency. Traffic scenarios vary greatly in complexity, from low-volume intersections to high-density junctions with highly dynamic flows. A fixed reasoning strategy is inadequate, as it leads to redundant computation in simple cases and insufficient analysis in complex ones, making it difficult to handle the diverse reasoning demands across traffic environments.

In response to these challenges, we propose CoLLMLight, the first cooperative LLM agent framework for network-wide traffic signal control. This framework is supported by an asynchronous cooperative decision architecture. To ensure effective cooperation, the architecture performs spatiotemporal-aware cooperative reasoning, conducting step-by-step analysis of the spatiotemporal context around the target intersection and deriving cooperative control guidance. To meet real-time requirements, the reasoning process is executed asynchronously and its results are cached. The decision module then retrieves the latest cooperative guidance as contextual input and integrates it with real-time observations to quickly select the optimal signal. Moreover, to enhance both cooperation effectiveness and reasoning efficiency, we propose a cost-aware cooperation optimization strategy. It first fine-tunes the LLM through adaptive reasoning chain optimization, enabling it to adjust reasoning depth according to traffic complexity. The model is then further refined with reinforcement learning using tailored reward functions, which guide both reasoning and decision policies toward efficient reasoning and network-wide traffic optimization.

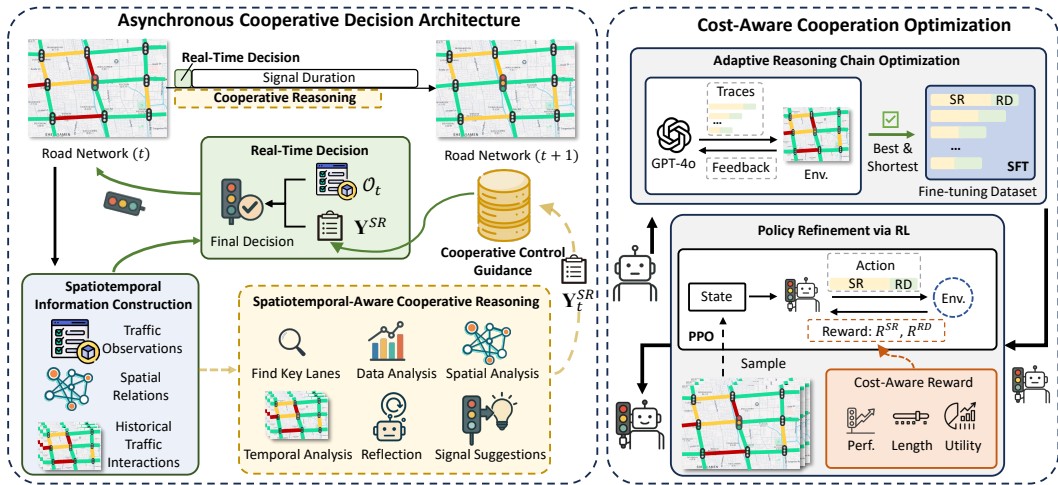

Figure 2: The overview of CoLLMLight framework.

Our contributions are summarized as follows: (1) We propose **CoLLMLight**, the first cooperative LLM agent framework for traffic signal control, enabling agents to achieve network-wide traffic optimization through inter-intersection cooperation. (2) We design an asynchronous cooperative decision architecture that supports real-time decision-making while enabling multi-step cooperative reasoning over rich spatiotemporal contexts. (3) We introduce a cost-aware cooperation optimization strategy that aligns reasoning depth with the demands of different traffic scenarios and improves both efficiency and cooperation quality. (4) Extensive evaluations on four real-world datasets demonstrate the effectiveness and superiority of CoLLMLight.

## 2 PROBLEM STATEMENT

In this section, we introduce key concepts related to traffic signal control and the formal problem statement. A more detailed illustration of the intersection layout, lane types, and signal phases is provided in the Appendix A.2.

**Definition 1 (Road Network)** *The road network is a directed graph composed of intersections $\mathcal{I}$ and lanes $\mathcal{L}$. Lanes can be categorized into three types: 1) go-through lanes ($\mathcal{L}_{go}$), 2) left-turn lanes ($\mathcal{L}_{left}$), 3) right-turn lanes ($\mathcal{L}_{right}$). Each lane connects two intersections and interacts with adjacent lanes at intersections.*

**Definition 2 (Traffic Signals)** *At each signal-switching time step, the agent assigned to the intersection selects a signal from the predefined signal set $\mathcal{A} = \{a_1, \ldots, a_m\}$. The traffic signal is represented as $a = set(\mathcal{L}_{allow})$, where $\mathcal{L}_{allow}$ is a group of allowed-to-go lanes without conflicting movements (i.e., a green light for $\mathcal{L}_{allow}$ and a red light for others).*

**Problem 1 (Cooperative LLM Agent for Network-Wide Traffic Signal Control)** *Consider a road network with multiple intersections, where each intersection $i \in \mathcal{I}$ is controlled by an LLM agent that follows a shared control policy $\pi$. At each signal-switching timestep $t$, the $i$-th agent receives: 1) traffic observations from its own and neighboring intersections $\mathcal{O}_t^i$; 2) spatial relations with nearby intersections $\mathcal{G}^i$; 3) historical traffic interactions involving itself and its neighbors $\mathcal{T}_t^i$. Based on these inputs, the agent selects the traffic signal action $a_t^i$ from the action space $\mathcal{A}^i$:*

$$a_t^i = \pi([\mathcal{O}_t^i, \mathcal{G}^i, \mathcal{T}_t^i], \mathcal{A}^i). \tag{1}$$

*The goal is to develop a cooperative LLM Agent with policy $\pi$ that maps traffic context to signal decisions, aiming to improve network-wide traffic efficiency, evaluated by metrics such as average queue length, travel time, and waiting time.*

## 3 METHODOLOGY

We present the overview of **CoLLMLight** in Figure 2, including: *1) Asynchronous Cooperative Decision Architecture.* This component enables effective multi-step cooperative reasoning while decoupling it from decision-making to ensure real-time signal control. At each timestep $t$, it executes two modules asynchronously: the spatiotemporal-aware cooperative reasoning module, which performs reasoning over historical traffic interactions and spatial relations and caches the results $\mathbf{Y}_t^{\text{SR}}$; and the real-time decision module, which selects the signal based on the current observation $\mathcal{O}_t$ and the latest cooperative control guidance $\mathbf{Y}^{\text{SR}}$. *2) Cost-Aware Cooperation Optimization.* This component fine-tunes the LLM to improve cooperation quality while ensuring reasoning efficiency. First, adaptive reasoning chain optimization collects reasoning traces of varying lengths by prompting GPT-4o in simulation, retaining only those that are both concise and effective for supervised fine-tuning. Then, reinforcement learning with cost-aware reward signals is applied to further refine reasoning and decision policies, promoting efficient reasoning and network-wide traffic performance.

### 3.1 ASYNCHRONOUS COOPERATIVE DECISION ARCHITECTURE

To enable effective multi-step cooperative reasoning over the spatiotemporal context and ensure real-time signal selection, we design an asynchronous decision architecture. At each timestep $t$, the agent first collects spatiotemporal information from its own intersection and neighboring collaborators. It then asynchronously triggers two modules: spatiotemporal-aware cooperative reasoning (SR) and real-time decision (RD).

#### 3.1.1 SPATIOTEMPORAL INFORMATION CONSTRUCTION

At each decision timestep $t$, we collect traffic condition features for each lane $l$:

$$\mathbf{o}_t^l = \left[ n_l^{\text{queue}}, \ n_l^{\text{move}}, \ \tau_l, \ \rho_l \right], \tag{2}$$

where $n_l^{\text{queue}}$ is the number of queued vehicles, $n_l^{\text{move}}$ is the number of moving vehicles, $\tau_l$ is the average waiting time, and $\rho_l \in [0,1]$ is the occupancy of lane $l$. These lane-level features are aggregated to construct the traffic observation of intersection $i$ at timestep $t$:

$$\mathcal{O}_t^i = \left\{ \mathbf{o}_t^l \mid l \in \mathcal{L}^i \right\}, \tag{3}$$

where $\mathcal{L}^i$ denotes the set of lanes connected to intersection $i$ and its neighbors. We represent the spatial relations around intersection $i$ using a directed subgraph $\mathcal{G}^i = (\mathcal{V}^i, \mathcal{L}^i)$, where $\mathcal{V}^i$ includes $i$ and its neighboring intersections. To capture temporal dynamics, we collect a sequence of historical observations and signal actions within a fixed time window $\Delta t$:

$$\mathcal{T}_t^i = \left\{ (\mathcal{O}_{t'}^i, \mathbf{a}_{t'}^i) \mid t - \Delta t < t' < t \right\}, \tag{4}$$

where $\mathbf{a}_{t'}^i$ denotes the signal decisions of both intersection $i$ and its neighboring intersections at timestep $t'$.

#### 3.1.2 SPATIOTEMPORAL-AWARE COOPERATIVE REASONING

Based on the constructed spatiotemporal context $(\mathcal{O}_t^i, \mathcal{G}^i, \mathcal{T}_t^i)$, each agent performs multi-step reasoning at timestep $t$ to analyze traffic dynamics and generate cooperative signal suggestions that consider both local efficiency and network-wide optimization.

Specifically, the spatiotemporal context is transformed into a human-readable prompt, which is then processed by the LLM:

$$\mathbf{Y}_t^{\text{SR},i} = f_{\text{LLM}}^{\text{SR}} \left( \text{Prompt}(\mathcal{O}_t^i, \mathcal{G}^i, \mathcal{T}_t^i) \right), \tag{5}$$

where $\mathbf{Y}_t^{\text{SR},i}$ represents the reasoning outcomes. In this process, the agent adaptively activates the necessary reasoning steps to comprehensively understand the traffic situation. The *critical lane identification* step determines which movements contribute most to current or emerging congestion and assigns them higher priority. The *spatial interaction analysis* step examines lane-level dependencies between the target intersection and its neighbors to capture how their traffic conditions mutually influence one another. The *temporal pattern analysis* step identifies short-term traffic evolution trends,

enabling the agent to anticipate near-future congestion. The *reflection* step evaluates the outcomes of recent signal decisions to avoid repeating ineffective or inefficient control actions. Finally, these analyses collectively enable the agent to generate cooperative guidance that provides conditional signal recommendations informed by predicted short-term traffic patterns. The reasoning output produced by SR is cached as textual context for later real-time decisions, and the model's ability to adaptively select the necessary reasoning steps is further refined during the optimization stage.

### 3.1.3 REAL-TIME DECISION

At each decision timestep $t$, the agent selects the signal phase based on the current traffic observation $\mathcal{O}_t^i$ and the cached reasoning result $\mathbf{Y}^{\text{SR},i}$:

$$a_t^i = f_{\text{LLM}}^{\text{RD}}\left(\text{Prompt}(\mathcal{O}_t^i, \mathbf{Y}^{\text{SR},i})\right), \tag{6}$$

where $a_t^i$ denotes the selected signal for intersection $i$ at timestep $t$. In this process, the agent only needs brief analysis to make decisions, thereby enabling real-time response. Notably, the agent does not simply follow the previously reasoned cooperative suggestions $\mathbf{Y}^{\text{SR},i}$; instead, it leverages them as contextual guidance and makes judgments based on the latest observation $\mathcal{O}_t^i$. When sudden traffic condition changes occur and the reasoning results derived from historical data offer limited benefit, the agent adaptively prioritizes real-time observations to ensure effective decisions.

## 3.2 COST-AWARE COOPERATION OPTIMIZATION

To improve cooperation quality while ensuring cost-effective reasoning, we propose cost-aware cooperation optimization, which consists of two stages: adaptive reasoning chain optimization and policy refinement via reinforcement learning.

### 3.2.1 ADAPTIVE REASONING CHAIN OPTIMIZATION

This stage fine-tunes the LLM agent to ensure format-compliant outputs while establishing ability to adaptively adjust reasoning depth to traffic conditions. We begin by sampling diverse traffic scenarios from the simulator. For each scenario, we prompt GPT-4o to generate multiple SR traces of varying lengths, each followed by a corresponding RD output. Among these combinations, we identify the SR–RD pair where the shortest SR trace is sufficient to support an RD output that achieves the best long-term network-wide traffic performance (*i.e.*, lowest average queue length). The selected SR and RD samples are then compiled into a supervised fine-tuning dataset $\mathcal{D}_{\text{SFT}}$.

We then fine-tune the LLM by minimizing the negative log-likelihood over this dataset:

$$\mathcal{L}_{\text{SFT}}(\theta) = - \sum_{(\mathbf{X}, \mathbf{Y}^*) \in \mathcal{D}_{\text{SFT}}} \sum_{w=1}^{|\mathbf{Y}^*|} \log P_{\pi_\theta}\left(y_w^* \mid \mathbf{X}, \mathbf{Y}_{<w}^*\right), \tag{7}$$

where $\pi_\theta$ denotes the LLM policy parameterized by $\theta$, $\mathbf{X}$ is the input prompt (for either SR or RD), and $\mathbf{Y}^*$ is the corresponding target output.

### 3.2.2 POLICY REFINEMENT VIA REINFORCEMENT LEARNING

This stage employs RL to jointly refine the SR and RD modules through interactions with the traffic environment. To guide this process, we design specialized reward signals for the reasoning and decision stages.

The reward $R^{\text{RD}}$ is assigned to the real-time decision module by comparing the agent's selected signal $a$ with the long-term optimal signal $a^*$:

$$R^{\text{RD}} = \begin{cases} +1, & \text{if } a = a^* \\ -1, & \text{otherwise,} \end{cases} \tag{8}$$

where $a^*$ is determined by evaluating all candidate signal phases in simulation and selecting the one that minimizes the long-term average queue length.

The reward $R^{\text{SR}}$ balances reasoning utility and computational cost. It integrates the downstream RD reward $R^{\text{RD}}$, the reasoning length $L$, and a binary utility score $U \in \{0, 1\}$. The utility score $U$ is

assigned by the agent during the RD process, indicating whether the SR provides useful cooperative guidance for its decision. The final reward is defined as:

$$R^{\text{SR}} = R^{\text{RD}} \cdot \left[ \beta \left( 1 - \frac{L}{L_{\max}} \right) + (1 - \beta)U \right], \qquad (9)$$

where $L_{\max}$ denotes the maximum reasoning length and $\beta \in [0, 1]$ controls the trade-off between reasoning conciseness and quality.

We optimize the LLM using Proximal Policy Optimization (PPO) (Schulman et al., 2017). The PPO objective maximizes the following clipped surrogate:

$$J^{\text{PPO}}(\theta) = \hat{\mathbb{E}}_k \left[ \min \left( r_k(\theta) \hat{A}_k, \; \text{clip}(r_k(\theta), 1 - \epsilon, 1 + \epsilon) \hat{A}_k \right) \right], \qquad (10)$$

where $k$ indexes the sampled steps in the training trajectory, $r_k(\theta) = \frac{\pi_\theta(y_k|s_k)}{\pi_{\theta_{\text{old}}}(y_k|s_k)}$ is the probability ratio between the current policy and the previous policy for the generated output $y_k$ given state $s_k$, and $\hat{A}_k$ is the estimated advantage computed from the corresponding reward signal.

## 4 EXPERIMENTS

### 4.1 EXPERIMENTAL SETUP

**Datasets.** Our experiments were conducted on one synthetic dataset (Syn-Train) and four real-world traffic flow datasets (Jinan, Hangzhou, New York 1, New York 2) (Mei et al., 2024). Detailed information about these datasets is provided in Appendix A.3. Syn-Train is a dataset that we synthesized. In our experiments, all learning-based methods (RL-based, LLMLight, CoLLMLight) are trained on Syn-Train and evaluated for their zero-shot performance on the real-world datasets.

**Environment Settings.** We utilize CityFlow (Zhang et al., 2019), a widely used open-source simulator, for our experiments. Each intersection operates with four signal phases: ETWT (east-west through), ELWL (east-west left-turn), NTST (north-south through), and NLSL (north-south left-turn). Each dataset simulates one hour. In the simulation, right-turn movements are allowed at all times. The green signal phase lasts for thirty seconds, followed by a three-second yellow phase and a two-second all-red phase (Zhang et al., 2022; Wei et al., 2019b; Lai et al., 2025).

**Baselines.** We compare our proposed method with two conventional transportation methods: Fixed-Time (Koonce et al., 2008) and MaxPressure (Varaiya, 2013); nine RL-based approaches: MPLight (Chen et al., 2020), AttendLight (Oroojlooy et al., 2020), PressLight (Wei et al., 2019a), CoLight (Wei et al., 2019b), Efficient-CoLight (Wu et al., 2021), Advanced-CoLight (Zhang et al., 2022), X-Light (Jiang et al., 2024), CosLight (Ruan et al., 2024), and DuaLight (Lu et al., 2024); and an SOTA LLM-based method: LLMLight (Lai et al., 2025). Additionally, we assess the performance of general LLMs integrated within our CoLLMLight framework, including Llama 3.1 (8B and 70B), Qwen 3 (8B and 32B), and the Deepseek-R1 distilled models (R1-8B and R1-32B).

**Evaluation Metrics.** We adopt *Average Travel Time* (ATT), *Average Waiting Time* (AWT), and *Average Queue Length* (AQL) (Wei et al., 2021; Zhang et al., 2022; Lai et al., 2025) as evaluation metrics to assess the performance of different traffic signal control methods. Specifically, ATT measures the average time taken for vehicles to travel from their origins to their destinations. AWT captures the average time vehicles spend waiting at intersections. AQL indicates the average number of vehicles queued at each intersection.

### 4.2 PERFORMANCE COMPARISON

To evaluate CoLLMLight against existing methods under unseen traffic scenarios, we adopt a zero-shot setting. All learning-based methods are fully trained on Syn-Train and directly tested on four real-world datasets. The results are reported in Table 1.

Compared to all existing baselines, CoLLMLight consistently achieves SOTA performance across all datasets and evaluation metrics, significantly outperforming both transportation-based and learning-based approaches. This demonstrates not only its strong generalization ability but also the effectiveness of its cooperative decision across diverse urban scenarios.

Table 1: Zero-shot performance comparison across different datasets (lower is better). Best results are shown in **bold**. Both CoLLMLight-8B and LLMLight-8B are finetuned from Llama3.1-8B

| Method | New York 1 | | | New York 2 | | | Jinan | | | Hangzhou | | |
|--------|-----|-----|-----|-----|-----|-----|-----|-----|-----|-----|-----|-----|
| | ATT | AWT | AQL | ATT | AWT | AQL | ATT | AWT | AQL | ATT | AWT | AQL |
| **Transportation Methods** | | | | | | | | | | | | |
| FixedTime | 1535.6 | 290.1 | 3173.6 | 1771.7 | 428.7 | 4523.1 | 441.2 | 66.7 | 294.1 | 616.0 | 74.0 | 301.3 |
| MaxPressure | 1223.3 | 153.3 | 2410.2 | 1566.8 | 255.5 | 3936.5 | 273.2 | 38.2 | 106.6 | 325.3 | 49.6 | 68.9 |
| **RL Methods** | | | | | | | | | | | | |
| MPLight | 1492.2 | 262.2 | 3054.1 | 1711.0 | 364.6 | 4324.5 | 482.2 | 84.6 | 340.6 | 496.2 | 64.7 | 197.5 |
| AttendLight | 1267.7 | 292.8 | 2557.2 | 1755.3 | 496.8 | 4571.8 | 310.2 | 63.4 | 144.5 | 327.2 | 72.9 | 69.0 |
| PressLight | 1687.2 | 508.4 | 3542.7 | 1894.0 | 489.7 | 4820.7 | 410.1 | 139.2 | 270.4 | 600.0 | 259.2 | 301.1 |
| CoLight | 1427.9 | 246.7 | 2923.0 | 1753.5 | 457.7 | 4517.6 | 474.4 | 90.8 | 333.1 | 530.2 | 90.3 | 228.1 |
| E-CoLight | 1266.0 | 281.3 | 2664.2 | 1645.7 | 437.3 | 4327.5 | 844.8 | 489.9 | 799.2 | 874.5 | 466.8 | 540.8 |
| A-CoLight | 1037.9 | 185.5 | 1845.8 | 1428.9 | 359.8 | 3665.6 | 392.9 | 144.7 | 242.9 | 428.0 | 235.9 | 156.5 |
| CosLight | 1851.4 | 761.8 | 3225.1 | 1986.9 | 816.3 | 4025.8 | 875.9 | 752.5 | 890.3 | 848.7 | 696.5 | 557.6 |
| X-Light | 1608.7 | 281.6 | 3313.9 | 1815.6 | 368.8 | 4635.9 | 661.1 | 117.3 | 548.6 | 695.8 | 117.5 | 352.6 |
| DuaLight | 1102.3 | 337.3 | 2026.1 | 1487.7 | 467.5 | 3952.3 | 425.5 | 230.7 | 289.7 | 558.5 | 435.1 | 277.3 |
| **LLMLight Framework** | | | | | | | | | | | | |
| Llama3.1-8B | 1289.2 | 549.1 | 2551.4 | 1618.3 | 582.1 | 4190.5 | 332.9 | 100.3 | 168.0 | 355.0 | 110.7 | 91.0 |
| LLMLight-8B | 1187.4 | 143.1 | 2297.9 | 1599.4 | 388.7 | 4074.6 | 268.6 | 40.5 | 102.2 | 312.0 | 39.5 | 58.1 |
| **CoLLMLight Framework (Ours)** | | | | | | | | | | | | |
| Llama3.1-8B | 1196.5 | 242.3 | 2342.8 | 1540.7 | 388.7 | 3917.7 | 281.3 | 44.7 | 114.3 | 333.8 | 45.6 | 73.5 |
| Qwen3-8B | 1109.2 | 116.6 | 2134.1 | 1435.9 | 245.0 | 3666.7 | 268.8 | 37.3 | 100.8 | 311.1 | 40.3 | 56.5 |
| Qwen3-32B | 1040.8 | 114.6 | 1939.4 | 1381.2 | 237.5 | 3510.3 | 277.1 | 38.5 | 110.6 | 314.8 | 40.9 | 59.5 |
| R1-8B | 1253.9 | 189.1 | 2455.4 | 1551.7 | 321.2 | 3910.7 | 329.8 | 76.9 | 167.9 | 372.8 | 91.5 | 105.8 |
| R1-32B | 1117.2 | 159.3 | 2050.2 | 1375.4 | 246.3 | 3463.1 | 272.6 | 37.7 | 105.3 | 310.9 | 37.1 | 56.3 |
| Llama3.1-70B | 1082.6 | 101.3 | 2060.5 | 1427.6 | 245.0 | 3644.8 | 269.2 | 36.1 | 101.9 | 313.7 | 34.8 | 58.5 |
| **CoLLMLight-8B** | **1000.4** | **90.5** | **1816.8** | **1345.1** | **167.6** | **3394.2** | **267.9** | **34.8** | **100.4** | **308.5** | **33.5** | **54.9** |

Among prior methods, MaxPressure shows relatively better generalization, benefiting from its queue-difference pressure heuristic. In contrast, RL-based models suffer from performance inconsistencies when applied to unseen networks. For example, Advanced-CoLight performs better in New York, while AttendLight excels in Jinan and Hangzhou, despite all models being trained solely on the synthetic network. This suggests that these methods are inherently biased toward certain structural or traffic patterns, resulting in uneven performance across test scenarios with varying characteristics, revealing their limited robustness under zero-shot evaluation. Conventional multi-agent RL methods represent traffic states as latent vectors and learn cooperation implicitly, which tends to anchor the resulting policies to the traffic patterns observed during training. In contrast, CoLLMLight interprets traffic states through explicit semantic reasoning, enabling more generalizable cooperative signal control. Additionally, we report the supervised performance of RL-based methods and CoLLMLight on Syn-Train in the Appendix A.6.

Moreover, LLMLight, which adopts an independent TSC agent with a localized signal selection strategy, performs competitively on simpler networks such as Jinan and Hangzhou. However, its performance drops in more complex settings like New York, where densely distributed intersections increase the need for cooperative signal control. This highlights the limitation of purely local decision-making. In contrast, CoLLMLight employs asynchronous spatiotemporal reasoning across intersections, allowing agents to anticipate upstream and downstream traffic before making real-time decisions, thereby achieving consistent advantages across all networks.

We further evaluate general-purpose LLMs within the CoLLMLight framework. While larger models (32B/70B) achieve reasonable results due to their capacity, they are consistently outperformed by our optimized CoLLMLight-8B. This confirms the benefit of our optimization strategy over scaling.

## 4.3 INFERENCE LATENCY ANALYSIS

We evaluate the inference latency of SR and RD modules across 8B-scale LLMs, as shown in Figure 3. Larger models (32B, 70B) are excluded due to higher latency and cost. RD time reflects real-time decision latency, crucial for timely switching. LLMLight lacks an SR module and performs all reasoning and decisions in RD, leading to higher latency than most models—except R1, which conducts excessive reasoning. In contrast, models in our framework achieve low RD latency.

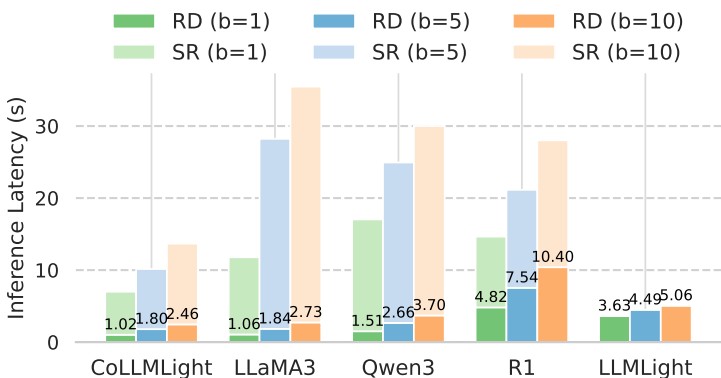

Figure 3: Inference time comparison of 8B-scale LLMs over batch sizes $b \in \{1, 5, 10\}$.

Table 2: Impact of SR design on average travel time (in seconds) across four datasets.

| Method | New York 1 | New York 2 | Jinan | Hangzhou |
|---|---|---|---|---|
| Async SR | **1000.4** | **1345.1** | 267.9 | 308.5 |
| Sync SR | 1005.7 | 1351.0 | **266.9** | **307.2** |
| w/o SR | 1155.1 | 1477.3 | 267.1 | 308.8 |

Although SR takes longer than RD due to multi-step reasoning, it runs asynchronously and does not delay signal decision. Among these, CoLLMLight achieves the lowest latency across all batch sizes and maintains RD latency below the typical yellow-light duration (3-5s) (Koonce et al., 2008), ensuring real-time applicability. Moreover, it shows the lowest SR latency, validating the effectiveness of our cost-aware cooperation optimization in reducing reasoning overhead.

## 4.4 ABLATION STUDY

### 4.4.1 IMPACT OF SPATIOTEMPORAL-AWARE COOPERATIVE REASONING (SR)

We compare three CoLLMLight variants to evaluate SR: (1) *Async SR*, where SR runs asynchronously and RD uses cached SR output $\mathbf{Y}^{SR}$ and the latest observation $\mathcal{O}_t^i$ for decision-making; (2) *Sync SR*, where SR runs synchronously and RD uses $\mathbf{Y}_t^{SR}$ and $\mathcal{O}_t^i$; (3) *w/o SR*, where SR is removed and RD relies only on $\mathcal{O}_t^i$. As shown in Table 2, removing SR increases average travel time, especially in New York, confirming its importance for cooperation. Async SR performs comparably to Sync SR across datasets, showing that asynchronous design avoids latency while preserving reasoning effectiveness. Although Async SR omits $t$-step observations, RD combines its output with the latest $\mathcal{O}_t^i$, maintaining decision quality. Performance differences are smaller in Jinan and Hangzhou, whose sparse topologies make agents rely more on real-time observations than historical patterns, reducing SR's contribution. Despite this, Async SR achieves comparable results, suggesting that the RD module effectively adapts to latest traffic conditions. This highlights RD's flexibility and its ability to operate reliably even when cooperative reasoning offers limited guidance.

### 4.4.2 EFFECT OF OPTIMIZATION STAGES

We conduct an ablation study on the two stages of our cost-aware cooperation optimization: adaptive reasoning chain optimization (AR) and policy refinement (PR). Table 3 compares four variants: the full model (Ours), w/o AR, w/o PR, and w/o Both. The results show that removing either AR or PR negatively impacts performance, confirming the effectiveness of both modules, with AR playing a dominant role. Specifically, removing AR significantly increases the reasoning chain length, with approximately 52% more tokens on the New York 1 dataset and 55% more tokens on Jinan. It also leads to a notable rise in ATT. This highlights the necessity of AR for adaptive reasoning. On this basis, PR further enhances both traffic performance and reasoning efficiency.

Table 3: Ablation study of optimization stages on average travel time (ATT, in seconds) and reasoning length (Token) of the SR module across New York 1 and Jinan.

| Variant | New York 1 | | Jinan | |
|---|---|---|---|---|
| | ATT ↓ | Token ↓ | ATT ↓ | Token ↓ |
| Ours | **1000.4** | **484.2** | **267.9** | **691.3** |
| w/o AR | 1066.3 | 738.5 | 270.3 | 1072.6 |
| w/o PR | 1034.2 | 543.9 | 269.8 | 759.2 |
| w/o Both | 1196.5 | 809.2 | 281.3 | 1198.5 |

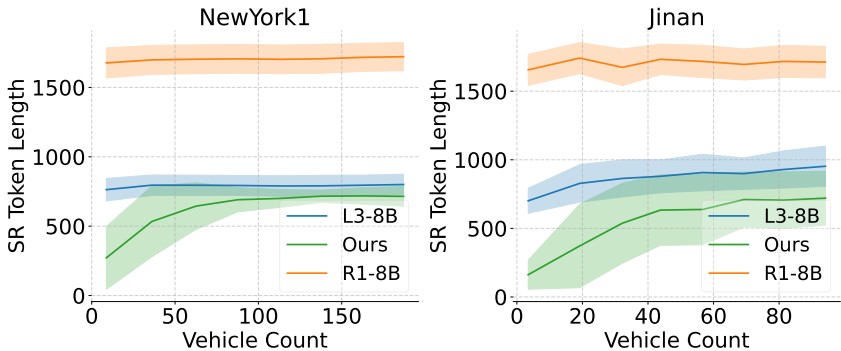

Figure 4: SR token length across traffic conditions with different vehicle counts.

## 4.5 REASONING BEHAVIOR ANALYSIS

To evaluate reasoning adaptiveness and efficiency, we analyze the token length of spatiotemporal-aware cooperative reasoning under varying traffic conditions. Specifically, we measure the number of vehicles each agent observes during SR, including both its own and neighboring intersections. The results are shown in Figure 4, where we compare our method with two baselines: Llama3.1-8B (L3-8B) and R1-8B. We observe that CoLLMLight produces shorter reasoning chains under light traffic. When faced with higher vehicle counts, which imply more complex traffic conditions, CoLLMLight dynamically increases its reasoning effort, resulting in increased SR token length. This behavior demonstrates the model's ability to adaptively adjust its reasoning depth according to traffic complexity, validating the effectiveness of our cost-aware cooperation optimization strategy. In contrast, L3-8B and R1-8B adopt fixed-length reasoning across all traffic conditions, with consistently longer reasoning chains than CoLLMLight.

## 4.6 ROBUSTNESS EVALUATION

To evaluate the robustness of CoLLMLight under network instability, we introduce two perturbed scenarios. The first is *Stale SR*, where the SR reasoning is delayed by one additional timestep with a probability of 50%. The second is *Communication Failure*, where the traffic state of a randomly selected neighboring intersection fails to be communicated, also with a probability of 50%. Table 4 reports the ATT performance for New York 1 and New York 2. For reference, we also report the normal setting, where SR reasoning finishes within one signal timestep and all collaborators communicate successfully. Across both perturbed scenarios, CoLLMLight exhibits only minor performance degradation, indicating that the asynchronous cooperative decision architecture remains stable. This robustness is supported by the design of the two modules. The SR module reasons over historical traffic states, which reduces sensitivity to intermittent communication issues. The RD module makes decisions based on real-time observations rather than blindly following potentially outdated SR guidance.

Table 4: Robustness test of CoLLMLight (ATT, lower is better).

| Scenario / Dataset | New York 1 | New York 2 |
|---|---|---|
| Normal | 1000.4 | 1345.1 |
| Stale SR | 1017.7 | 1380.4 |
| Communication Failure | 1028.1 | 1381.7 |

## 5 RELATED WORK

Traffic signal control presents a long-standing challenge in intelligent transportation systems, with approaches evolving from transportation-based methods to RL-based methods, and more recently, to LLM-based agents. Transportation-based methods such as FixedTime (Koonce et al., 2008), which employs fixed cycle lengths and phase allocations, and MaxPressure (Varaiya, 2013), which reduces congestion by minimizing queue imbalance across traffic movements, were widely used in early traffic control systems. However, they rely on manually designed rules and often struggle to adapt in complex and dynamic traffic environments. RL-based methods significantly advanced traffic signal control by introducing innovative neural architectures and refined state representations. On the architectural side, FRAP (Zheng et al., 2019) models dynamic phase-level interactions; CoLight (Wei et al., 2019b) leverages graph attention networks (Veličković et al., 2017) for inter-intersection coordination; and CosLight (Ruan et al., 2024) utilizes a multilayer perceptron to construct a collaborator matrix across intersections. In parallel, state representation techniques have become increasingly sophisticated. PressLight (Wei et al., 2019a) proposes pressure-based features; Efficient-XLight (Wu et al., 2021) extends this idea by computing pressure in a lane-to-lane manner for finer granularity; and Advanced-XLight (Zhang et al., 2022) further improves performance by incorporating effective vehicle counts. Although RL-based methods have demonstrated strong performance, their cooperative behavior is encoded implicitly within neural architectures. Moreover, these models often struggle to transfer their learned coordination strategies to unseen road networks, limiting their generalization capability. More recently, LLM-based TSC agents have emerged as a promising new paradigm. Approaches such as LLMLight (Lai et al., 2025) and VLMLight (Wang et al., 2025) exhibit strong generalization and human-like reasoning abilities. However, existing LLM-based controllers primarily focus on local decision-making and lack explicit mechanisms for inter-intersection cooperation, which is essential for achieving network-level coordination. In contrast, CoLLMLight adopts a multi-agent architecture and enables explicit semantic cooperative reasoning over spatiotemporal information, while still maintaining real-time responsiveness.

## 6 CONCLUSION

In this paper, we propose CoLLMLight, a cooperative LLM agent framework for network-wide traffic signal control. Specifically, we design an asynchronous cooperative decision architecture. This architecture executes spatiotemporal-aware cooperative reasoning and the real-time decision module asynchronously, enabling effective inter-intersection cooperation while ensuring real-time responsiveness. Moreover, we propose a cost-aware cooperation optimization strategy with two stages. The first stage applies adaptive reasoning optimization, allowing the LLM to adjust its reasoning depth according to traffic conditions. The second stage refines the policy using reward signals that promote both decision quality and reasoning efficiency. Extensive experiments on four real-world datasets demonstrate the superiority of CoLLMLight.

## ACKNOWLEDGMENTS

This work was supported by the National Natural Science Foundation of China (Grant No. 62572417 and No. 92370204), the National Key R&D Program of China (Grant No. 2023YFF0725004), and the CCF-DiDi GAIA Collaborative Research Funds.

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

Table 5: Statistics of datasets.

| Dataset | No. Inter. | No. Veh. | Arrival rate (vehicles/5 min) | | | |
|---|---|---|---|---|---|---|
| | | | Mean | Std | Max | Min |
| Syn-Train | 16 | 8000 | 666.6 | 32.1 | 735 | 621 |
| Jinan | 12 | 4365 | 362.8 | 74.8 | 493 | 236 |
| Hangzhou | 16 | 2983 | 247.6 | 40.4 | 332 | 211 |
| New York 1 | 196 | 11058 | 849.6 | 174.0 | 964 | 382 |
| New York 2 | 196 | 16337 | 1255.7 | 264.8 | 1440 | 475 |

# A APPENDIX

## A.1 THE USE OF LARGE LANGUAGE MODELS (LLMS)

LLMs are a core component of our methodology. We propose CoLLMLight, the first cooperative LLM agent framework for network-wide traffic signal control. In addition, LLMs are employed during manuscript preparation to refine grammar and improve clarity.

## A.2 SETTINGS OF TRAFFIC SIGNAL CONTROL

We present the most commonly used settings in Figure 5.

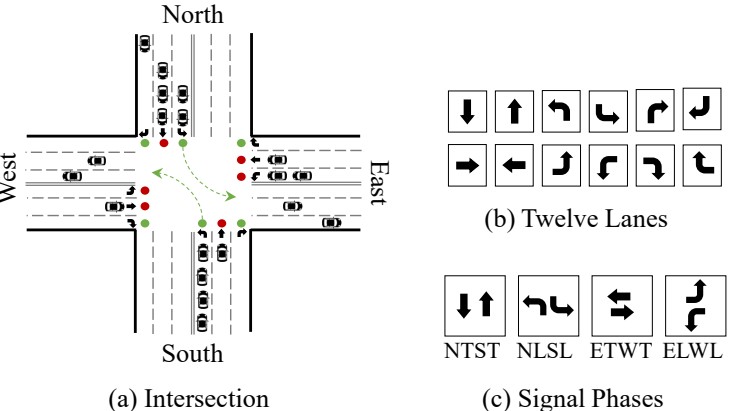

(a) Intersection

(b) Twelve Lanes

(c) Signal Phases

Figure 5: An illustration of intersections, lanes, and signal phases. The signal phases include ETWT (east-west through movement), ELWL (left-turn from east and west), NTST (north-south through movement), and NLSL (left-turn from north and south)

## A.3 DATASET DESCRIPTION

- **Jinan**: A dataset from the Dongfeng sub-district in Jinan, China, consisting of 12 intersections. Each intersection features two 400-meter roads (east-west) and two 800-meter roads (north-south).

- **Hangzhou**: A dataset from the Gudang sub-district in Hangzhou, China, comprising 16 intersections, with each intersection featuring two 800-meter roads (east-west) and two 600-meter roads (north-south).

- **New York**: Collected in Manhattan's Upper East Side using taxi trip data, this extensive dataset encompasses 196 intersections. It includes two large-scale traffic flow datasets from different periods.

- **Syn-Train**: A synthetic dataset based on a $4 \times 4$ grid network, where each road segment is 300 meters long. It serves as the primary training set for learning-based traffic signal control methods in our experiments.

Comprehensive statistics for these datasets are presented in Table 5. Figure 7 presents the road-network layouts of the real-world datasets.

Figure 6 shows the empirical distribution of per-intersection vehicle counts observed in our simulations. Although the New York network is far denser than Jinan and Hangzhou, most of its intersections still face light demand, while a small fraction experience extremely heavy congestion (over 200 vehicles). This wide variance underscores the importance of adaptive reasoning.

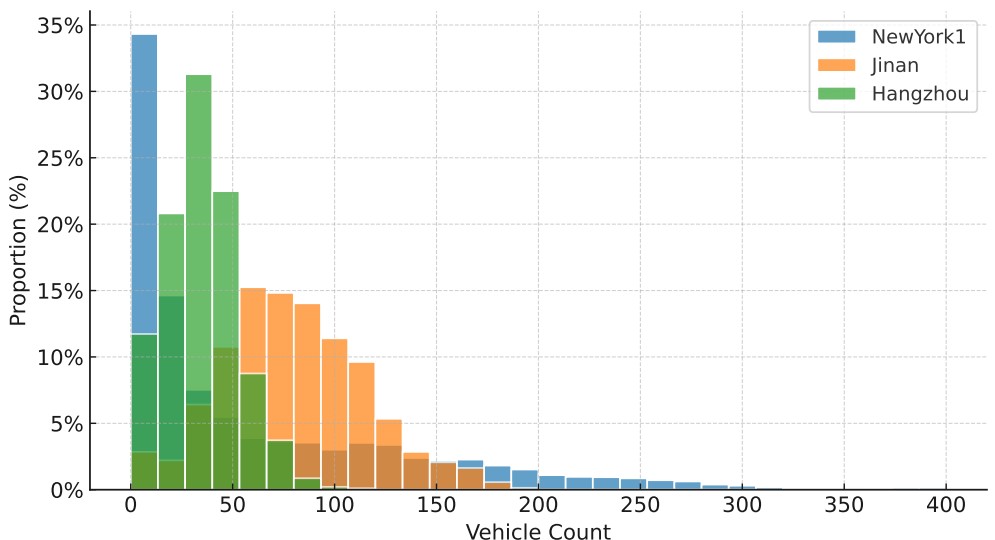

Figure 6: Distribution of per-agent observed vehicle counts across datasets. While most intersections in New York experience low traffic volumes, a notable fraction face extremely high traffic.

## A.4    COMPARED METHODS

- **FixedTime** (Koonce et al., 2008): A traditional signal control method using static, predefined cycle lengths and phase times.
- **MaxPressure** (Varaiya, 2013): A SOTA transportation method selecting signal phases based on queue length pressure between upstream and downstream intersections.
- **MPLight** (Chen et al., 2020): It utilizes pressure as both observation and reward, based on the FRAP.
- **AttendLight** (Oroojlooy et al., 2020): It uses attention mechanisms to predict phase transitions and construct observation features.
- **PressLight** (Wei et al., 2019a): Applies Deep Reinforcement Learning (DRL) to optimize intersection pressure.
- **CoLight** (Wei et al., 2019b): Employs graph attention network (GAT) for inter-intersection communication.
- **Efficient-CoLight** (Wu et al., 2021): Enhances CoLight by integrating efficient pressure observations.
- **Advanced-CoLight** (Zhang et al., 2022): A SOTA MARL-based method that enhances CoLight by integrating advanced traffic state features, including pressure and effective running vehicles.
- **X-Light** (Jiang et al., 2024): A meta-MARL framework that leverages a transformer-on-transformer architecture to extract scenario-shared and scenario-specific knowledge for cross-city transfer.
- **CosLight** (Ruan et al., 2024): A collaboration-aware method that jointly optimizes collaborator selection and signal decision-making to enhance inter-intersection cooperation.

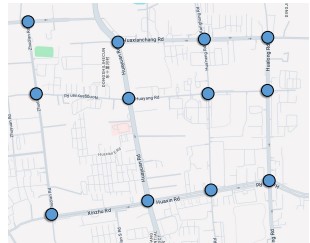 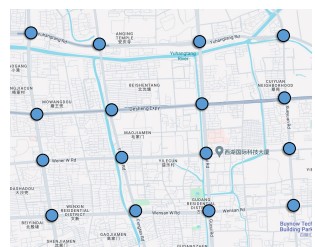 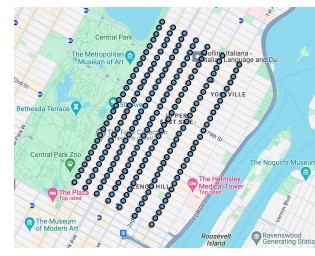

(a) Dongfeng, Jinan, China.    (b) Gudang, Hangzhou, China.    (c) Upper East Side, Manhattan, New York, US.

Figure 7: The road network layouts of the Jinan, Hangzhou, and New York datasets (Lai et al., 2025).

Table 6: Comparative Performance of Learning-based Methods at Syn-Train

| Method | ATT | AWT | AQL |
|---|---|---|---|
| MPLight | 979.03 | 504.66 | 568.29 |
| AttendLight | 753.09 | 152.85 | 502.38 |
| PressLight | 715.45 | 403.20 | **371.90** |
| CosLight | 949.97 | 646.26 | 502.24 |
| X-Light | 989.18 | 183.25 | 525.46 |
| DuaLight | **621.63** | 531.31 | 451.08 |
| CoLight | 826.44 | 260.96 | 492.87 |
| E–CoLight | 716.40 | 361.47 | 521.49 |
| A–CoLight | 671.60 | 348.82 | 491.97 |
| LLMLight | 1256.93 | 411.15 | 790.06 |
| CoLLMLight | 868.81 | **110.02** | 579.41 |

- **DuaLight** (Lu et al., 2024): A dual-branch MARL approach that separately models scenario-specific and scenario-shared knowledge for improved adaptability across diverse traffic conditions.

- **LLMLight** (Lai et al., 2025): A SOTA LLM TSC agent that employs a reasoning process mimicking human-like intuition to optimize traffic management.

## A.5 IMPLEMENTATION DETAILS

We conducted our experiments on Linux servers equipped with two A800 GPUs. All RL methods were trained with consistent hyperparameters: a learning rate of $1 \times 10^{-3}$, a replay buffer size of 12,000, a sample size of 3,000, and a hidden layer size of 20, for 100 training epochs. For LLMs, the temperature parameter is set to 0.1. We perform LoRA fine-tuning on Llama 3.1 (8B) to obtain LLMLight and CoLLMLight, using a LoRA rank of 8, a scaling factor $\alpha = 16$, and a learning rate of $1 \times 10^{-4}$. In CoLLMLight, the historical observation window is set to $\Delta t = 5$, and the reward balancing coefficient in Equation 9 is set to $\beta = 0.5$.

## A.6 SUPERVISED PERFORMANCE COMPARISON

We present the supervised performance on Syn-Train in Table 6. From these results, we observe that DuaLight and Advanced-CoLight (A-Light) achieve the lowest ATT, while Presslight yields the lowest AQL. In contrast, our method, CoLLMLight, attains the lowest AWT. The strong performance of RL-based methods on Syn-Train can be attributed to their effective training on this synthetic environment. However, when applied to real-world datasets, both RL-based methods exhibit significant performance degradation. In contrast, CoLLMLight maintains superior performance, demonstrating its strong generalization capability.

## A.7 CASE STUDY

To demonstrate the superiority of CoLLMLight, we conduct a case study on the New York-1 dataset. This scenario represents a typical spillback dilemma, where a locally congested lane is blocked downstream, forcing agents to balance the urgency of local queue discharge against network-level stability. As shown in Table 7, we compare the spatiotemporal reasoning (SR) and real-time decisions (RD) of the baseline models with those of CoLLMLight.

In this scenario, persistent activation of the NTST phase in previous timesteps has caused severe congestion in the downstream lane of NT, while WT is under strong upstream pressure and its downstream section remains relatively free. From the SR–RD outputs, we observe that LLaMA-3.1 8B does not sufficiently account for the downstream blockage of NT and still releasing NTST, which would further worsen the spillback.

Both LLaMA-3.1 70B and CoLLMLight-8B select the ETWT phase. However, CoLLMLight demonstrates more comprehensive reasoning: its SR module not only identifies both upstream pressure and downstream blockage, but also performs prediction and enumerates plausible candidate strategies. This structured anticipation enables the RD module to make a fast and accurate decision, ultimately selecting ETWT to relieve WT's upstream neighbors and stabilize the network.

## A.8 VARIANCE AND STATISTICAL SIGNIFICANCE ANALYSIS

Each model is evaluated using five random seeds (11, 42, 1024, 2025, 20), and the corresponding performance variance is summarized in Table 8. To further assess robustness, we conduct Welch's t-tests comparing all baselines against CoLLMLight-8B.

Across nearly all datasets and metrics, CoLLMLight-8B achieves highly significant improvements ($p < 0.001$). Only a few isolated exceptions fall outside this range. Specifically, non-significant differences ($p \geq 0.05$) are observed for A-CoLight on New York 1 (AQL), Qwen3-8B on Jinan (AQL), and R1-32B on Hangzhou (AQL). A small number of moderately significant outcomes ($0.01 \leq p < 0.05$) appear for R1-32B on New York 2 (AQL) and Llama3.1-70B on New York 1 (AWT). Additionally, R1-32B shows one intermediate significance case ($0.001 \leq p < 0.01$) on Hangzhou (ATT).

Aside from these rare cases, all remaining comparisons consistently demonstrate that CoLLMLight-8B outperforms alternative methods with strong and statistically significant margins.

## A.9 COMMUNICATION MECHANISM

Figure 8 illustrates the neighborhood structure in a simple $3 \times 3$ grid network. In our problem setting, the neighborhood radius is defined based on signal-controlled lanes rather than purely topological one-hop distance. In real-world urban networks, some adjacent intersections include uncontrolled right-turn lanes, and such lanes allow the dependency to extend to the next signal-controlled intersection located at the corresponding two-hop position. Consequently, the neighborhood of an agent consists of all directly adjacent intersections and a small number of two-hop intersections that become reachable when a neighboring connection passes through an uncontrolled lane. For example, in a simple $3 \times 3$ grid, if the target intersection is at the center, its communication neighborhood includes the surrounding eight intersections because each is reachable through controlled lanes or through connections that extend across uncontrolled right-turns. The lane set used for observation follows the same dependency rules. It includes all controlled lanes of the target intersection, as well as lanes from neighboring intersections that maintain direct upstream or downstream relationships with the target intersection's approaches or exits, including those reached through an uncontrolled connector.

## A.10 COLLMLIGHT PSEUDOCODE

The procedure in Algorithm 1 describes how CoLLMLight makes signal decisions at each timestep.

Table 7: Case Study at Timestep 115. The table provides a concise summary of the SR–RD process, while a complete prompt–response example is included in Appendix A.15.

| Case Description (Step 114 → 115) |
|---|
| **Historical Information (Brief Summary, T109 → T114):** T109 (NTST): NT has a queue of 36; WT fully congested; upstream saturated. T110 (NTST): NT still 36; WT remains fully congested. T111 (NTST): NT drops slightly (36→30); WT still 37; NT downstream overloaded. T112 (NTST): NT drops to 12; WT still 37. T114 (ETWT): WT slightly relieved (37→36); NT rebounds to 28; upstream still saturated. |
| **Real-Time Observation (Brief Summary, T115):** At T115, both NT and WT remain highly congested. NT's downstream lanes are overloaded (Spillback Risk), while WT's downstream is free. Upstream pressure persists on all major approaches. |
| **Expert Action:** `ETWT`. *Rationale:* Prioritize WT to clear upstream congestion. Do not release NT to prevent downstream spillback. |

| Model | Brief Summary of Spatiotemporal-aware Cooperative Reasoning (SR) | Real-time Decision (RD) |
|---|---|---|
| **Llama 3.1 8B** | NT and WT are the critical congested lanes, with spillback affecting their upstream approaches. The previous signal phase did not mitigate the buildup, and keeping the current phase will likely worsen delays. A phase change is needed: ETWT or NLSL are the most suitable options to relieve NT/WT pressure, though ETWT may introduce new congestion on the ET approach. | **Action:** `NTST` (*Wrong*). **Rationale:** The traffic state matches ATR predictions of NT/WT pressure and upstream spillback. Selecting NTST is the most suitable choice to maintain flow on the most critical approaches and aligns with the local priority ranking. ATR guidance directly supports this decision. |
| **Llama 3.1 70B** | WT and NT are the main congested lanes with spillback from multiple upstream approaches. Previous signals only partially relieved pressure, and congestion is expected to persist at the next timestep. Two feasible options emerge: ETWT to relieve WT or NTST to relieve NT. The appropriate choice depends on which lane's pressure is prioritized at timestep 115. | **Action:** `ETWT`. **Rationale:** Congestion on WT persists as predicted by ATR, and ETWT is the top-ranked option in the Local Signal Priority. Since WT's downstream lanes have sufficient capacity, activating ETWT is the appropriate choice to relieve WT effectively. |
| **CoLLMLight 8B** | NT and WT remain the critical lanes, both under heavy upstream pressure; releasing either would help prevent further spillback. The previous phase partially eased NT but left WT congested, indicating additional release is needed. Without intervention, queues on both lanes—especially their upstream feeders—are likely to worsen. A conditional strategy is recommended: use NTST if NT's downstream path is clear, otherwise select ETWT to relieve WT and its critical upstream neighbors. | **Action:** `ETWT`. **Rationale:** WT remains heavily congested with critical upstream pressure, and releasing it would prevent further spillback. ATR identifies ETWT as appropriate given that WT's downstream is clear, and this matches the Local Signal Priority ranking, where ETWT offers the greatest waiting-time reduction. |

## A.11 SPATIOTEMPORAL ABLATION STUDY

We conducted ablation experiments on the NewYork dataset to examine whether CoLLMLight relies on spatial and temporal context during decision-making. In each setting, specific components of the input prompts were removed to isolate their contribution. Table 9 reports the ATT results for New York 1 and New York 2. The results show that removing either temporal or spatial information leads to a noticeable drop in performance, and the degradation becomes more severe when both are removed. In particular, the absence of spatial context produces a substantial decline in accuracy,

Table 8: Zero-shot performance comparison across different datasets (lower is better). Best results are shown in bold. The lower block shows LLMs evaluated with the CoLLMLight agent framework.

| Models | New York 1 | | | New York 2 | | | Jinan | | | Hangzhou | | |
|---|---|---|---|---|---|---|---|---|---|---|---|---|
| | ATT | AWT | AQL | ATT | AWT | AQL | ATT | AWT | AQL | ATT | AWT | AQL |
| FixedTime | 1535.6 (±0) | 290.1 (±0) | 3173.6 (±0) | 1771.7 (±0) | 428.7 (±0) | 4523.1 (±0) | 441.2 (±0) | 66.7 (±0) | 294.1 (±0) | 616.0 (±0) | 74.0 (±0) | 301.3 (±0) |
| MaxPressure | 1223.3 (±0) | 153.3 (±0) | 2410.2 (±0) | 1566.8 (±0) | 255.5 (±0) | 3936.5 (±0) | 273.2 (±0) | 38.2 (±0) | 106.6 (±0) | 325.3 (±0) | 49.6 (±0) | 68.9 (±0) |
| MPLight | 1492.2 (±42.80) | 262.2 (±8.75) | 3054.1 (±87.46) | 1711.0 (±27.11) | 364.6 (±7.43) | 4324.5 (±62.47) | 482.2 (±35.14) | 84.6 (±5.60) | 340.6 (±39.49) | 496.2 (±21.28) | 64.7 (±2.87) | 197.5 (±15.16) |
| AttendLight | 1267.7 (±20.07) | 292.8 (±16.46) | 2557.2 (±22.74) | 1755.3 (±25.52) | 496.8 (±9.12) | 4571.8 (±36.32) | 310.2 (±6.24) | 63.4 (±1.42) | 144.5 (±5.01) | 327.2 (±2.27) | 72.9 (±3.66) | 69.0 (±0.71) |
| PressLight | 1687.2 (±21.25) | 508.4 (±13.92) | 3542.7 (±48.34) | 1894.0 (±14.96) | 489.7 (±38.56) | 4820.7 (±36.13) | 410.1 (±15.32) | 139.2 (±9.21) | 270.4 (±18.75) | 600.0 (±9.41) | 259.2 (±6.28) | 301.1 (±5.91) |
| CoLight | 1427.9 (±39.05) | 246.7 (±11.44) | 2923.0 (±71.00) | 1753.5 (±43.98) | 457.7 (±27.27) | 4517.6 (±108.36) | 474.4 (±6.82) | 90.8 (±4.13) | 333.1 (±5.68) | 530.2 (±17.32) | 90.3 (±5.22) | 228.1 (±12.78) |
| E-CoLight | 1266.0 (±16.98) | 281.3 (±11.36) | 2664.2 (±67.19) | 1645.7 (±20.93) | 437.3 (±9.17) | 4327.5 (±25.50) | 844.8 (±99.41) | 489.9 (±66.87) | 799.2 (±76.91) | 874.5 (±38.01) | 466.8 (±57.91) | 540.8 (±38.11) |
| A-CoLight | 1037.9 (±17.13) | 185.5 (±16.75) | 1845.8 (±93.74) | 1428.9 (±17.68) | 359.8 (±2.10) | 3665.6 (±49.52) | 392.9 (±3.43) | 144.7 (±3.56) | 242.9 (±3.81) | 428.0 (±5.73) | 235.9 (±6.38) | 156.5 (±6.21) |
| CosLight | 1851.4 (±7.60) | 761.8 (±12.36) | 3225.1 (±17.77) | 1986.9 (±15.47) | 816.3 (±14.08) | 4025.8 (±43.08) | 875.9 (±4.25) | 752.5 (±8.69) | 890.3 (±4.77) | 848.7 (±5.21) | 696.5 (±15.72) | 557.6 (±5.18) |
| X-Light | 1608.7 (±11.27) | 281.6 (±6.01) | 3313.9 (±25.88) | 1815.6 (±12.78) | 368.8 (±8.16) | 4635.9 (±9.36) | 661.1 (±46.84) | 117.3 (±9.83) | 548.6 (±57.19) | 695.8 (±25.45) | 117.5 (±11.46) | 352.6 (±16.71) |
| DualLight | 1102.3 (±14.79) | 337.3 (±4.45) | 2026.1 (±57.64) | 1487.7 (±14.53) | 467.5 (±17.84) | 3952.3 (±54.67) | 425.5 (±13.95) | 230.7 (±25.77) | 289.7 (±21.02) | 558.5 (±39.15) | 435.1 (±36.22) | 277.3 (±35.26) |
| LLMLight-8B | 1187.4 (±12.19) | 143.1 (±5.17) | 2297.9 (±3.25) | 1599.4 (±6.65) | 388.7 (±14.44) | 4074.6 (±10.55) | 268.6 (±1.41) | 40.5 (±1.06) | 102.2 (±1.58) | 312.0 (±0.62) | 39.5 (±0.63) | 58.1 (±0.54) |
| Llama3.1-8B | 1196.5 (±23.08) | 242.3 (±70.69) | 2342.8 (±46.57) | 1540.7 (±12.76) | 388.7 (±1.49) | 3917.7 (±33.17) | 281.3 (±2.06) | 44.7 (±1.71) | 114.3 (±1.88) | 333.8 (±0.09) | 45.6 (±0.18) | 73.5 (±0.12) |
| Qwen3-8B | 1109.2 (±20.26) | 116.6 (±3.33) | 2134.1 (±45.96) | 1435.9 (±6.48) | 245.0 (±6.06) | 3666.7 (±34.55) | 268.8 (±0.30) | 37.3 (±0.04) | 100.8 (±0.10) | 311.1 (±2.89) | 40.3 (±0.04) | 56.5 (±1.33) |
| R1-8B | 1253.9 (±1.25) | 189.1 (±16.38) | 2455.4 (±59.03) | 1551.7 (±8.30) | 321.2 (±6.06) | 3910.7 (±49.97) | 329.8 (±0.37) | 76.9 (±0.66) | 167.9 (±0.67) | 372.8 (±1.53) | 91.5 (±0.09) | 105.8 (±2.91) |
| Qwen3-32B | 1040.8 (±10.59) | 114.6 (±2.37) | 1939.4 (±27.05) | 1381.2 (±3.10) | 237.5 (±4.72) | 3510.3 (±32.01) | 277.1 (±0.33) | 38.5 (±0.10) | 110.6 (±0.54) | 314.8 (±1.25) | 40.9 (±0.04) | 59.5 (±1.58) |
| R1-32B | 1117.2 (±5.45) | 159.3 (±11.63) | 2050.2 (±34.41) | 1375.4 (±6.21) | 246.3 (±4.38) | 3463.1 (±54.24) | 272.6 (±0.30) | 37.7 (±0.58) | 105.3 (±0.23) | 310.9 (±1.65) | 37.1 (±0.04) | 56.3 (±1.45) |
| Llama3.1-70B | 1082.6 (±4.33) | 101.3 (±9.00) | 2060.5 (±12.99) | 1427.6 (±8.37) | 245.0 (±4.79) | 3644.8 (±21.06) | 269.2 (±0.28) | 36.1 (±0.11) | 101.9 (±0.15) | 313.7 (±2.12) | 34.8 (±0.02) | 58.5 (±1.59) |
| **CoLLMLight-8B** | **1000.4 (±8.67)** | **90.5 (±3.66)** | **1816.8 (±28.83)** | **1345.1 (±5.13)** | **167.6 (±4.96)** | **3394.2 (±46.24)** | **267.9 (±0.37)** | **34.8 (±0.37)** | **100.4 (±0.68)** | **308.5 (±1.12)** | **33.5 (±0.02)** | **54.9 (±0.95)** |

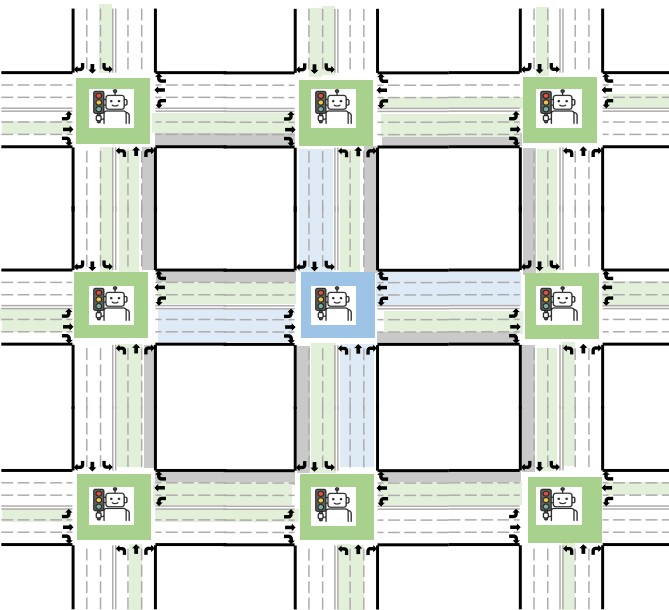

Figure 8: Communication neighborhood of an agent (blue) in a $3 \times 3$ grid network. The eight surrounding intersections (green) serve as its cooperative partners. Blue lanes denote the agent's controlled lanes, green lanes represent upstream or downstream lanes from neighboring intersections, and gray lanes indicate uncontrolled right-turn lanes.

indicating that the agent makes active use of information from neighboring intersections rather than behaving as an isolated controller.

Table 9: Ablation study results (ATT, lower is better).

| Method | New York 1 | New York 2 |
|---|---|---|
| **CoLLMLight (Full)** | **1000.4** | **1345.1** |
| w/o Temporal Context | 1155.1 | 1477.3 |
| w/o Spatial Context | 1112.6 | 1443.5 |
| w/o Both | 1168.7 | 1542.2 |

## A.12 HUMAN EVALUATION

To assess whether CoLLMLight's decisions are interpretable and safe, we conducted an expert inspection study with four transportation researchers experienced in traffic-signal control. The evalua-

---

**Algorithm 1: CoLLMLight Procedure for Intersection $i$ at Timestep $t$**

---

**Input:** Traffic observation $\mathcal{O}_t^i$, spatial relations $\mathcal{G}^i$, historical traffic interactions $\mathcal{T}_t^i$, previous cooperative guidance $Y_{\text{cache}}^{\text{SR},i}$ (if available)

**Output:** Signal action $a_t^i$, new cooperative guidance $Y_t^{\text{SR},i}$

1: **Asynchronous Execution:** Initiate the following two processes (RD and SR) in parallel:
2: **Process 1: Real-Time Decision (RD)**
3: Construct decision prompt $D_t^i$ based on current observation $\mathcal{O}_t^i$ and cached guidance $Y_{\text{cache}}^{\text{SR},i}$.
4: Query RD module to select signal phase: $a_t^i \leftarrow f_{LLM}^{RD}(D_t^i)$.
5: Execute action $a_t^i$ at intersection $i$.
6: **Process 2: Spatiotemporal-Aware Cooperative Reasoning (SR)**
7: Construct reasoning context $C_t^i$ based on $\mathcal{O}_t^i$, $\mathcal{G}^i$, and $\mathcal{T}_t^i$.
8: Query SR module with $C_t^i$ to **adaptively select and execute necessary reasoning steps** (skipping unnecessary ones in simple scenarios), such as:
9:    • Identify critical lanes (local and neighboring).
10:    • Analyze spatial interactions (upstream/downstream dependencies).
11:    • Analyze temporal patterns from history $\mathcal{T}_t^i$.
12:    • Reflect on previous decisions.
13: Generate cooperative control guidance: $Y_t^{\text{SR},i} \leftarrow f_{LLM}^{\text{SR}}(C_t^i)$.
14: $Y_{\text{cache}}^{\text{SR},i} \leftarrow Y_t^{\text{SR},i}$

---

tion was performed on 100 sampled SR-RD outputs from CoLLMLight-8B. Each sample was rated on a 1–5 scale across five dimensions: interpretability, internal consistency, decision soundness, reasoning validity, and risk-awareness. Table 10 reports the mean and standard deviation of the scores. The results indicate that CoLLMLight provides clear and coherent reasoning traces, reliably identifies potential safety risks such as downstream blockage, and selects signal phases consistent with established traffic-engineering principles.

Table 10: Human evaluation results.

| Metric | Mean $\pm$ Std |
|---|---|
| Interpretability | $4.78 \pm 0.17$ |
| Consistency | $4.59 \pm 0.26$ |
| Decision soundness | $4.71 \pm 0.12$ |
| Reasoning validity | $4.74 \pm 0.14$ |
| Risk-awareness | $4.83 \pm 0.11$ |
| Overall | $4.73 \pm 0.13$ |

## A.13 EXPERIMENT ON LARGE ROAD NETWORKS

CoLLMLight is designed to maintain efficient communication and reasoning as the network grows, since its overhead scales linearly with the number of upstream and downstream collaborators rather than with the size of the entire road network. We evaluate CoLLMLight on a large Manhattan road network consisting of 1176 intersections and an hourly demand of 46,541 vehicles. Following the same protocol as in our main experiments, all learning-based methods are trained on the Syn-Train dataset and evaluated zero-shot on this unseen large-scale network. Table 11 reports the ATT, AWT, and AQL results. CoLLMLight-8B achieves the best performance across all three metrics, with a consistently larger improvement margin compared to conventional MARL approaches. This suggests that the semantic reasoning mechanism remains effective and robust even under highly complex urban traffic conditions.

## A.14 FEASIBILITY OF REAL-WORLD DEPLOYMENT

The real-time feasibility of CoLLMLight is well within the capabilities of modern traffic-signal infrastructures. The communication payload is lightweight, as each agent transmits less than 1 KB

Table 11: Zero-shot performance comparison on the large Manhattan road network (lower is better).

| Method | ATT | AWT | AQL |
|---|---|---|---|
| FixedTime | 1901.82 | 885.05 | 14891.94 |
| MaxPressure | 1165.76 | 119.38 | 9223.00 |
| MPLight | 1392.39 | 189.82 | 12013.66 |
| AdvancedCoLight | 1195.48 | 333.31 | 9395.88 |
| CoLight | 1380.24 | 207.46 | 11872.34 |
| LLMLight-8B | 1111.76 | 144.98 | 8257.82 |
| **CoLLMLight-8B** | **1008.10** | **86.28** | **6902.18** |

of structured numerical lane features per signal interval, resulting in minimal bandwidth requirements. Although our experiments do not explicitly model communication latency, deployed intelligent transportation systems already achieve delays that are sufficient for real-time coordination; for example, the FHWA feasibility study (Rayamajhi et al., 2020) reports that Dedicated Short-Range Communications can achieve point-to-point delays below 2 ms. The inference time of CoLLMLight meets real-time operational needs as well. As shown in Figure 3, the RD module exhibits low latency with average delays of 1.02, 1.80, and 2.46 seconds when jointly processing batches of 1, 5, and 10 intersections on a $2 \times$A800 setup, which remains safely below typical yellow-light intervals of 3 to 6 seconds (McGee Sr et al., 2012). Deployment cost is also modest when the system is served through an LLM provider using API access. Current reference prices for 8B-scale models are low, with DeepInfra listing approximately $0.02 per million input tokens and $0.03 per million output tokens. Based on our measured token usage in the New York experiments (SR: 2331.69 input and 484.20 output tokens; RD: 1403.95 input and 109.69 output tokens; 2880 calls per module per day), the estimated daily cost is roughly $0.27 per intersection. API inference prices have been decreasing over time as model architectures and serving systems improve, suggesting that long-term operational costs will continue to decline (Xiao et al., 2025).

## A.15 PROMPT AND RESPONSE EXAMPLES

We provide representative prompt–response examples. In these prompts, "advance traffic reasoning" refers to spatiotemporal-aware cooperative reasoning. Additionally, we instruct the decision module to output the utility score corresponding to the cooperative reasoning result.

**Prompt: System Prompt**

1   You are a traffic signal control expert responsible for managing a
      four-way intersection within a larger road network. Your
      objective is to optimize both local and network-wide traffic
      flow and safety by selecting appropriate signal phases at each
      decision point. You must consider not only the traffic condition
      at your intersection, but also the influence on upstream and
      downstream intersections.

2

3   Your decision-making follows a two-stage process:

4

5   1. **Advance Traffic Reasoning:**

6      Based on current and historical traffic observations, you should
        conduct sufficient but necessary analysis to understand the
        key lanes, congestion risks, and propagation effects. Your
        goal is to provide a well-grounded recommendation and
        justification that can support the next-stage decision.

7

8   2. **Reactive Action:**

9      Given the real-time observation at timestep t and the prior
        reasoning, you must promptly determine the optimal signal
        phase. If significant changes are detected compared to
        previous reasoning, you should adapt your decision
        accordingly.

10

11  Follow all safety principles and ensure that each decision balances
      local urgency and global impact.

12

13  ## Background Context

14  An intersection has 12 lanes: [NL, NT, NR, SL, ST, SR, EL, ET, ER,
      WL, WT, WR]. Each lane is labeled by direction and movement: N
      for north, S for south, E for east, W for west, L for left turn,
       T for through, and R for right turn. For instance, ET stands
      for the East Through lane, where traffic moves straight ahead
      from east to west. WL is the West Left-turn lane, where traffic
      turns left from west to south. Right turns are always allowed.
      There are four signal options: [ETWT, NTST, ELWL, NLSL]. For
      example, ETWT indicates the release of both the ET and WT lanes.
      The signal phase duration is set to thirty seconds.

15

16  ## Note:

17  For each Lane X, when considering activating it, keep these in mind
      :

18     - NEVER let the occupancy of X's downstream lanes be close to
        100% at any risk, as it will cause severe congestion, this
        is FIRST PRIORITY.

19     - If the upstream or downstream information of X isn't
        mentioned, it means that they are in a good state with low
        occupancy.

20     - You MUST consider how much the occupancy of X's downstream
        lanes will increase upon releasing lane X.

21     - You MUST delay the release of X if its downstream has a high
        occupancy rate.

22     - If there are many high-occupancy lanes upstream of X and X's
        occupancy is not low, you MUST consider releasing X so as to
        help upstream lanes release.

23     - You can't keep a lane waiting for too long. You MUST release
        the lane with excessive waiting time when the downstream
        condition allows.

**Prompt: Spatiotemporal-Aware Cooperative Reasoning - 1**

```
1  You are now in the **Advance Traffic Reasoning** phase (at time
       step 114).
2  Your goal is to perform deep but adaptive reasoning based on
       historical and current traffic conditions. This reasoning will
       guide the signal decision for the next time step 115.
3  ---
4  ## Historical Observation
5  - Lanes include both signal-controlled lanes at the current
       intersection (e.g., NL, NT, SL, ST, EL, ET, WL, WT) and upstream
       /downstream lanes from neighboring intersections (e.g., SL's
       upstream lane (4, ST)). **Only lanes with vehicles are shown.**
6  - Values are shown as: value or value(+change from previous
       timestep).
7
8  Timestep 109 signal: NTST
9  Timestep 110 traffic states:
10 |Lane|Cars Input|Cars Output|Queued Cars|Moving Cars|Average
       Waiting Time (mins)|Occupancy (%)|
11 |NT|0|0|36(+0)|0(+0)|5.63(+0.5)|109.09(+0.0)|
12 |WT|0|0|37(+0)|0(+0)|1.49(+0.5)|112.12(+0.0)|
13 |NT's upstream lane (58, NT)|0|0|36(+0)|0(+0)|4.32(+0.5)
       |109.09(+0.0)|
14 |NT's downstream lane (28, ET)|3|10|0(-15)|21(+8)|0.0(-1.19)
       |63.64(-21.21)|
15 |NT's downstream lane (56, NT)|0|0|1(+0)|0(+0)|5.78(+0.5)
       |3.03(+0.0)|
16 |NL's upstream lane (58, NT)|0|0|36(+0)|0(+0)|4.32(+0.5)
       |109.09(+0.0)|
17 |NL's upstream lane (59, NT)|0|0|35(+0)|0(+0)|3.65(+0.5)
       |106.06(+0.0)|
18 |WT's upstream lane (28, ST)|1|0|7(+0)|1(+1)|2.19(+0.5)
       |24.24(+3.03)|
19 |WT's upstream lane (56, SL)|0|0|38(+0)|0(+0)|1.14(+0.5)
       |115.15(+0.0)|
20 |WT's upstream lane (29, WT)|5|0|36(+13)|0(-8)|0.6(+0.36)
       |109.09(+15.15)|
21 |WT's upstream lane (29, NL)|0|0|14(+1)|0(-1)|2.04(+0.38)
       |42.42(+0.0)|
22
23 Timestep 110 signal: NTST
24 Timestep 111 traffic states:
25 |Lane|Cars Input|Cars Output|Queued Cars|Moving Cars|Average
       Waiting Time (mins)|Occupancy (%)|
26 |NT|0|0|36(+0)|0(+0)|6.13(+0.5)|109.09(+0.0)|
27 |WT|0|0|37(+0)|0(+0)|1.99(+0.5)|112.12(+0.0)|
28 |NT's upstream lane (58, NT)|0|0|36(+0)|0(+0)|4.82(+0.5)
       |109.09(+0.0)|
29 |NT's downstream lane (28, ET)|8|0|20(+20)|9(-12)|0.25(+0.25)
       |87.88(+24.24)|
30 |NT's downstream lane (56, NT)|0|0|1(+0)|0(+0)|6.28(+0.5)
       |3.03(+0.0)|
31 |NT's downstream lane (28, EL)|1|0|0(+0)|1(+1)|0.0(+0.0)
       |3.03(+3.03)|
32 |NL's upstream lane (58, NT)|0|0|36(+0)|0(+0)|4.82(+0.5)
       |109.09(+0.0)|
33 |NL's upstream lane (59, NT)|0|0|35(+0)|0(+0)|4.15(+0.5)
       |106.06(+0.0)|
34 |WT's upstream lane (28, ST)|1|0|8(+1)|1(+0)|2.38(+0.19)
       |27.27(+3.03)|
35 |WT's upstream lane (56, SL)|0|0|38(+0)|0(+0)|1.64(+0.5)
       |115.15(+0.0)|
36 |WT's upstream lane (29, WT)|0|0|36(+0)|0(+0)|1.1(+0.5)
       |109.09(+0.0)|
37 |WT's upstream lane (29, NL)|0|0|14(+0)|0(+0)|2.54(+0.5)
       |42.42(+0.0)|
```

---

**Prompt: Spatiotemporal-Aware Cooperative Reasoning - 2**

```
 1  Timestep 111 signal: NTST
 2  Timestep 112 traffic states:
 3  |Lane|Cars Input|Cars Output|Queued Cars|Moving Cars|Average
       Waiting Time (mins)|Occupancy (%)|
 4  |NT|0|2|30(-6)|4(+4)|5.88(-0.25)|103.03(-6.06)|
 5  |WT|0|0|37(+0)|0(+0)|2.49(+0.5)|112.12(+0.0)|
 6  |NT's upstream lane (58, NT)|0|0|36(+0)|0(+0)|5.32(+0.5)
       |109.09(+0.0)|
 7  |NT's downstream lane (28, ET)|7|0|36(+16)|0(-9)|0.56(+0.31)
       |109.09(+21.21)|
 8  |NT's downstream lane (56, NT)|0|0|1(+0)|0(+0)|6.78(+0.5)
       |3.03(+0.0)|
 9  |NT's downstream lane (28, EL)|1|0|1(+1)|1(+0)|0.33(+0.33)
       |6.06(+3.03)|
10  |NL's upstream lane (58, NT)|0|0|36(+0)|0(+0)|5.32(+0.5)
       |109.09(+0.0)|
11  |NL's upstream lane (59, NT)|0|0|35(+0)|0(+0)|4.65(+0.5)
       |106.06(+0.0)|
12  |WT's upstream lane (28, ST)|1|9|0(-8)|1(+0)|0.0(-2.38)
       |3.03(-24.24)|
13  |WT's upstream lane (56, SL)|0|8|17(-21)|13(+13)|1.91(+0.27)
       |90.91(-24.24)|
14  |WT's upstream lane (29, WT)|0|0|36(+0)|0(+0)|1.6(+0.5)
       |109.09(+0.0)|
15  |WT's upstream lane (29, NL)|0|0|14(+0)|0(+0)|3.04(+0.5)
       |42.42(+0.0)|
16
17  Timestep 112 signal: NTST
18  Timestep 113 traffic states:
19  |Lane|Cars Input|Cars Output|Queued Cars|Moving Cars|Average
       Waiting Time (mins)|Occupancy (%)|
20  |NT|0|6|12(-18)|16(+12)|1.1(-4.78)|84.85(-18.18)|
21  |WT|0|0|37(+0)|0(+0)|2.99(+0.5)|112.12(+0.0)|
22  |NT's upstream lane (58, NT)|0|0|36(+0)|0(+0)|5.82(+0.5)
       |109.09(+0.0)|
23  |NT's downstream lane (28, ET)|1|8|17(-19)|12(+12)|0.8(+0.24)
       |87.88(-21.21)|
24  |NT's downstream lane (56, NT)|3|0|1(+0)|3(+3)|7.28(+0.5)
       |12.12(+9.09)|
25  |NT's downstream lane (28, EL)|0|0|2(+1)|0(-1)|0.52(+0.19)
       |6.06(+0.0)|
26  |NL's upstream lane (58, NT)|0|0|36(+0)|0(+0)|5.82(+0.5)
       |109.09(+0.0)|
27  |NL's upstream lane (59, NT)|0|0|35(+0)|0(+0)|5.15(+0.5)
       |106.06(+0.0)|
28  |WT's upstream lane (28, ST)|1|0|1(+1)|1(+0)|0.08(+0.08)
       |6.06(+3.03)|
29  |WT's upstream lane (56, SL)|2|1|20(+3)|11(-2)|0.22(-1.69)
       |93.94(+3.03)|
30  |WT's upstream lane (29, WT)|0|0|36(+0)|0(+0)|2.1(+0.5)
       |109.09(+0.0)|
31  |WT's upstream lane (29, NL)|0|0|14(+0)|0(+0)|3.54(+0.5)
       |42.42(+0.0)|
```

**Prompt: Spatiotemporal-Aware Cooperative Reasoning - 3**

```
1   Timestep 113 signal: ETWT
2   Timestep 114 traffic states:
3   |Lane|Cars Input|Cars Output|Queued Cars|Moving Cars|Average
        Waiting Time (mins)|Occupancy (%)|
4   |NT|5|0|28(+16)|5(-11)|0.28(-0.82)|100.0(+15.15)|
5   |WT|0|1|36(-1)|0(+0)|1.9(-1.09)|109.09(-3.03)|
6   |NT's upstream lane (58, NT)|0|5|23(-13)|8(+8)|4.01(-1.81)
        |93.94(-15.15)|
7   |NT's downstream lane (28, ET)|1|1|19(+2)|10(-2)|0.22(-0.58)
        |87.88(+0.0)|
8   |NT's downstream lane (56, NT)|0|0|4(+3)|0(-3)|2.13(-5.15)
        |12.12(+0.0)|
9   |NT's downstream lane (28, EL)|1|0|2(+0)|1(+1)|1.02(+0.5)
        |9.09(+3.03)|
10  |NL's upstream lane (58, NT)|0|5|23(-13)|8(+8)|4.01(-1.81)
        |93.94(-15.15)|
11  |NL's upstream lane (59, NT)|0|0|35(+0)|0(+0)|5.65(+0.5)
        |106.06(+0.0)|
12  |NL's downstream lane (85, WT)|1|0|0(+0)|1(+1)|0.0(+0.0)
        |3.03(+3.03)|
13  |WT's upstream lane (28, ST)|0|0|2(+1)|0(-1)|0.47(+0.39)|6.06(+0.0)
        |
14  |WT's upstream lane (56, SL)|7|0|38(+18)|0(-11)|0.53(+0.31)
        |115.15(+21.21)|
15  |WT's upstream lane (29, WT)|0|0|36(+0)|0(+0)|2.6(+0.5)
        |109.09(+0.0)|
16  |WT's upstream lane (29, NL)|0|0|14(+0)|0(+0)|4.04(+0.5)
        |42.42(+0.0)|
17  |WT's downstream lane (85, WT)|1|0|0(+0)|1(+1)|0.0(+0.0)
        |3.03(+3.03)|
18  ---
19  ## Current Observation
20  Timestep 114 signal: ETWT
21  ---
22  ## Instruction
23  Please proceed through the following steps:
24  ### Step 1: Identify Critical Lanes
25  Examine the current intersection and its upstream/downstream
        neighbors to identify critical local and neighboring lanes that
        are critical or congested and require attention; output as:
26  Local Critical Lanes: ...
27  Neighboring Critical Lanes: ...
28
29  ### Step 2: Perform Adaptive Reasoning
30  Select and apply only the **necessary reasoning steps** to support
        an effective and fast decision at the next time step 115.
31
32  Suggested reasoning topics include (but are not limited to):
33  - Local queue and waiting time analysis
34  - Upstream/downstream traffic influence
35  - Reflection on previous signal effectiveness
36  - Prediction of the traffic state at timestep 115
37  - Anticipated effects of activating specific signal phases (e.g.,
        NTST, ETWT, NLSL, ELWL) at 115
38
39  ### Step 3: Conditional Signal Suggestions
40  Provide one or more signal strategies that may be appropriate at
        time step 115.
41  For each **suggested signal**, specify the situations under which
        it would be effective, including reasoning and expected effects.
```

**Prompt: Spatiotemporal-Aware Cooperative Reasoning - 4**

```
1   Structure each strategy as:
2
3   - **Suggested Signal**: The signal phase that may be activated at
        time step 115
4      - **Applicable Conditions**: What kind of traffic pattern or
            development this signal is suitable for
5      - **Rationale**: Why this signal is an appropriate choice under
             those conditions
6      - **Expected Effect**: The anticipated outcome if this signal
          is activated
7
8   ### Step 4: Summary
9
10  ## Notes
11  - Your reasoning should be adaptive: longer and deeper if traffic
        conditions are complex, shorter if simple.
12  - You may skip Steps 2 and 3 if the scenario is simple\u2014for
        example, when there are no critical neighboring lanes.
13
14  ## Output Format
15  Your output should follow the structure below:
16
17  ## Identify Critical Lanes
18
19  ## Reasoning Topic 1
20
21  ## Reasoning Topic 2
22
23  ...
24
25  ## Conditional Signal Suggestions
26
27  ## Summary
```

---

**Response: Spatiotemporal-Aware Cooperative Reasoning - 1**

```
 1  ## Identify Critical Lanes
 2
 3  Local Critical Lanes: NT, WT
 4  Neighboring Critical Lanes: NT's upstream lane (58, NT), WT's
        upstream lane (56, SL), WT's upstream lane (29, WT)
 5
 6  ## Reasoning Topic 1: Local Queue and Waiting Time Analysis
 7
 8  - NT has a high queue of 28 vehicles and an occupancy of 100%,
        indicating significant congestion. The average waiting time has
        decreased to 0.28 minutes, suggesting some relief but still
        requires attention.
 9  - WT has a queue of 36 vehicles and an occupancy of 109.09%, with a
        waiting time of 1.9 minutes. This lane is also critical and
        needs to be addressed soon.
10
11  ## Reasoning Topic 2: Upstream/Downstream Traffic Influence
12
13  - NT's upstream lane (58, NT) has reduced its queue to 23 vehicles,
        but still has a high occupancy of 93.94%. This indicates that
        NT needs to be cleared to prevent upstream congestion.
14  - WT's upstream lane (56, SL) has a high queue of 38 vehicles and
        an occupancy of 115.15%, which is critical. Releasing WT could
        help alleviate this upstream congestion.
15  - WT's upstream lane (29, WT) has a queue of 36 vehicles and an
        occupancy of 109.09%, also critical. Releasing WT could help
        here as well.
16
17  ## Reasoning Topic 3: Reflection on Previous Signal Effectiveness
18
19  - The previous signal phase ETWT was effective in reducing the
        queue in NT's downstream lane (28, ET) and NT's upstream lane
        (58, NT). However, WT's queue remains high, indicating that
        further release of WT is necessary.
20
21  ## Reasoning Topic 4: Prediction of Traffic State at Timestep 115
22
23  - If NT is not released, the queue will likely increase,
        exacerbating upstream congestion.
24  - If WT is not released, the queue will remain high, and upstream
        congestion will worsen.
25
26  ## Conditional Signal Suggestions
27
28  - **Suggested Signal**: NTST
29    - **Applicable Conditions**: Suitable if NT's downstream lanes
          are not critical, allowing for safe release of NT without
          causing downstream congestion.
30    - **Rationale**: NT has a high queue and occupancy, and its
          upstream lane is critical. Releasing NT will help alleviate
          upstream congestion and prevent further buildup.
31    - **Expected Effect**: Reduction in NT's queue and occupancy,
          relief for upstream congestion, and improved flow through
          the intersection.
```

**Response: Spatiotemporal-Aware Cooperative Reasoning - 2**

```
1  - **Suggested Signal**: ETWT
2     - **Applicable Conditions**: Suitable if WT's downstream lanes
         are not critical, allowing for safe release of WT without
         causing downstream congestion.
3     - **Rationale**: WT has a high queue and occupancy, and its
         upstream lanes are critical. Releasing WT will help
         alleviate upstream congestion and prevent further buildup.
4     - **Expected Effect**: Reduction in WT's queue and occupancy,
         relief for upstream congestion, and improved flow through
         the intersection.
5
6  ## Summary
7
8  The current traffic conditions at the intersection indicate
      critical congestion in NT and WT lanes, with significant
      upstream congestion affecting these lanes. The NTST signal phase
       is recommended if NT's downstream lanes are clear, to alleviate
       upstream congestion and reduce NT's queue. Alternatively, the
      ETWT signal phase is recommended if WT's downstream lanes are
      clear, to alleviate upstream congestion and reduce WT's queue.
      Careful monitoring of downstream conditions is essential to
      prevent cascading congestion.
```

**Prompt: Real-Time Decision - 1**

```
1   You are now in the **Reactive Action** phase at time step 115.
2
3   Your goal is to quickly and accurately select the signal phase to
        activate at this moment, based on:
4   - The prior **Advance Traffic Reasoning (ATR)** conducted at time
        step 114
5   - The **Current Observation** at time step 115
6   - The **Local Signal Priority** ranking based on estimated waiting
        time reduction
7
8   ---
9
10  [Cooperative Reasoning Result]
11
12  ---
13
14  # Current Observation
15  - Lanes include both signal-controlled lanes at the current
        intersection (e.g., NL, NT, SL, ST, EL, ET, WL, WT) and upstream
        /downstream lanes from neighboring intersections (e.g., SL's
        upstream lane (4, ST)). **Only lanes with vehicles are shown.**
16  - Values are shown as: value or value(+change from previous
        timestep).
17
18  Timestep 114 signal: ETWT
19  Timestep 115 traffic states:
20  |Lane|Cars Input|Cars Output|Queued Cars|Moving Cars|Average
        Waiting Time (mins)|Occupancy (%)|
21  |NT|3|0|36(+8)|0(-5)|0.7(+0.42)|109.09(+9.09)|
22  |NL|1|0|0(+0)|1(+1)|0.0(+0.0)|3.03(+3.03)|
23  |WT|0|0|36(+0)|0(+0)|2.4(+0.5)|109.09(+0.0)|
24  |NT's upstream lane (58, NT)|2|3|18(-5)|12(+4)|0.11(-3.9)
        |90.91(-3.03)|
25  |NT's downstream lane (28, ET)|7|0|36(+17)|0(-10)|0.53(+0.31)
        |109.09(+21.21)|
26  |NT's downstream lane (56, NT)|0|0|4(+0)|0(+0)|2.63(+0.5)
        |12.12(+0.0)|
27  |NT's downstream lane (28, EL)|3|0|4(+2)|2(+1)|0.81(-0.21)
        |18.18(+9.09)|
28  |NL's upstream lane (58, NT)|2|3|18(-5)|12(+4)|0.11(-3.9)
        |90.91(-3.03)|
29  |NL's upstream lane (59, NT)|0|3|25(-10)|7(+7)|6.06(+0.41)
        |96.97(-9.09)|
30  |NL's downstream lane (85, WT)|0|0|1(+1)|0(-1)|0.33(+0.33)
        |3.03(+0.0)|
31  |WT's upstream lane (28, ST)|1|0|2(+0)|1(+1)|0.97(+0.5)|9.09(+3.03)
        |
32  |WT's upstream lane (56, SL)|0|7|19(-19)|12(+12)|0.78(+0.25)
        |93.94(-21.21)|
33  |WT's upstream lane (29, WT)|0|0|36(+0)|0(+0)|3.1(+0.5)
        |109.09(+0.0)|
34  |WT's upstream lane (29, NL)|1|0|15(+1)|0(+0)|4.24(+0.2)
        |45.45(+3.03)|
35  |WT's downstream lane (85, WT)|0|0|1(+1)|0(-1)|0.33(+0.33)
        |3.03(+0.0)|
```

**Prompt: Real-Time Decision - 2**

```
1   # Instruction
2
3   Please:
4
5   1. Compare the ATR suggestions with the current traffic state.
6   2. If the current situation aligns with ATR predictions, select the
         recommended signal accordingly.
7   3. If the current situation **deviates** from ATR, make necessary
         adjustments to respond to new patterns.
8   4. If **upstream/downstream influence is minimal or negligible**,
         directly select the top-ranked signal from the Local Signal
         Priority table.
9   5. Do not congest downstream lanes. When the occupancy of Lane X,
         plus that of any other X's downstream lanes, is close to 100%,
         you must delay the release of X.
10  6. Decide whether the ATR output was **actually useful** in your
         decision using the 'used_atr' flag:
11     - 'used_atr = 1': The ATR output was referenced and influenced
            the decision.
12     - 'used_atr = 0': The decision was made independently of the ATR
            output.
13
14  ---
15
16  # Output Format
17
18  Return your result in the following **structured JSON format**:
19
20  ```json
21  {
22    \"signal\": \"<Selected signal phase, e.g., ETWT, NTST, NLSL,
         ELWL>\",
23    \"reason\": \"<Brief justification: why this signal is chosen
         based on traffic observation, local signal ranking, and (if
         applicable) the ATR suggestion.>\",
24    \"used_atr\": <0 or 1>
25  }
26  ```
```

**Response: Real-Time Decision**

```
1   ```json
2   {
3     \"signal\": \"ETWT\",
4     \"reason\": \"The current observation shows that WT has a high
         queue and occupancy, and its upstream lanes are critical.
         Releasing WT will help alleviate upstream congestion and
         reduce WT's queue. The ATR suggested ETWT as a viable option
         if WT's downstream lanes are not critical, which aligns with
         the current state. Additionally, ETWT is the top-ranked signal
          in the Local Signal Priority table, offering the highest
         waiting time reduction.\",
5     \"used_atr\": 1
6   }
7   ```
```

