# OpenReview forum: "CoLLMLight: Cooperative Large Language Model Agents for Network-Wide Traffic Signal Control"
_ICLR.cc/2026/Conference — ICLR 2026 Poster_

### Official Review · Reviewer_TspF · 2025-10-30

**Soundness:** 3
**Presentation:** 3
**Contribution:** 2
**Rating:** 4
**Confidence:** 3

**Summary:**

This paper, CoLLMLight: Cooperative Large Language Model Agents for Network-Wide Traffic Signal Control, proposes a framework where multiple LLM-based agents collaboratively manage a network of intersections. The key ideas are an asynchronous cooperative decision architecture that separates reasoning from real-time control, and a cost-aware cooperation optimization mechanism that balances reasoning depth and computational efficiency. Experiments on several real-world traffic datasets show promising improvements over both reinforcement learning (RL) and LLM-based baselines.

**Strengths:**

Novel problem framing: The paper extends prior LLM-based traffic control works (e.g., LLMLight) to a cooperative multi-agent setting, which is a meaningful and realistic step toward network-level optimization.

Asynchronous architecture: Separating reasoning and decision modules to maintain real-time control is a practical and well-motivated design.

Cost-aware optimization idea: Introducing adaptive reasoning and reinforcement learning to regulate reasoning depth shows awareness of the token and latency constraints in LLM systems.

**Weaknesses:**

- The proposed method appears as an incremental extension of LLMLight with cooperation and token-efficiency modules. Both the asynchronous design and adaptive reasoning could be considered engineering optimizations rather than fundamentally new learning mechanisms.

- The cooperative reasoning process relies on textual prompting, but there is little analysis of why or how the LLM reasoning benefits coordination beyond the cached contextual hints. No ablation or visualization clarifies what the LLM is learning about spatial-temporal dependencies.

- Although the asynchronous module mitigates latency, the experiments rely on simulation rather than real-time control. It’s unclear whether the system can truly operate under realistic time constraints with communication and inference overhead among dozens of intersections.

- The zero-shot evaluation uses synthetic training and real-world testing but does not report variance or statistical significance.

- No human or expert evaluation validates the interpretability or safety of the LLM’s decisions.

- Comparison with strong graph-based MARL methods (e.g., multi-agent transformers) is missing.

- The paper devotes extensive space to performance tables and prompts but offers limited insights into why the model works, missing a deeper understanding of LLM-agent cooperation dynamics.

**Questions:**

Besides weakness, I also have the following questions:

How are LLM reasoning traces (SR) concretely represented and reused during decision making? Are they natural-language summaries or latent embeddings?

How is the asynchronous communication among intersections implemented? Do agents share text tokens, numeric signals, or both?

How robust is CoLLMLight to communication failures or stale reasoning caches in dynamic conditions?

---

> ### Author Response · Authors · 2025-11-24
> **Response to Reviewer TspF Part 1**
>
> We sincerely thank the reviewer for the detailed and constructive feedback.
> To address all concerns clearly, we provide point-by-point responses below.
>
> ---
> # W1: Clarification on the novelty and contributions of CoLLMLight
> > The proposed method appears as an incremental extension of LLMLight with cooperation and token-efficiency modules. Both the asynchronous design and adaptive reasoning could be considered engineering optimizations rather than fundamentally new learning mechanisms.
>
> We sincerely appreciate the reviewer’s feedback regarding the novelty and contributions of our work. We respectfully clarify that CoLLMLight introduces three fundamental contributions that address critical challenges— effective inter-intersection cooperation, real-time feasibility, and efficiency—inherent in applying LLMs to complex traffic systems.
>
> **1. From Local Optimization to Network-wide Optimization:**
> LLMLight pioneers the use of LLMs for traffic control but operates as independent agents that make decisions based solely on local observations. This independence fundamentally limits its ability to address network-wide traffic efficiency optimization challenges.
> In contrast, CoLLMLight establishes the **first cooperative LLM agent framework for traffic signal control**. By enabling agents to communicate and reason about spatiotemporal traffic contexts, it empowers agents to consider inter-intersection cooperation and achieve network-wide optimization that is unattainable for independent agents.
>
> **2. Bridging High Reasoning Latency and Real-Time Control:**
> Directly applying LLMs to real-time control is theoretically unsound due to the significant latency incurred by complex reasoning.
> Our contribution is designing the *Asynchronous Cooperative Decision Architecture*, which asynchronously activates "System 2 thinking [1]" (i.e., spatiotemporal-aware cooperative reasoning) and "System 1 execution" (i.e., real-time decision-making). It ensures that high-latency semantic intelligence acts as a strategic guide without blocking real-time actuation, offering a valuable blueprint for deploying LLM agents in time-sensitive traffic systems.
>
> **3. Balancing Token Efficiency with Control Effectiveness:**
> We introduce *Cost-Aware Cooperation Optimization*, which achieves two key objectives:
>
> * **Adaptive Reasoning:**
>   It optimizes the length and depth of reasoning traces, ensuring the agent generates concise yet sufficient reasoning for the current traffic state.
>
> * **Cooperation Effectiveness:**
>   Through integration with RL, the semantic reasoning is grounded in environmental feedback, ensuring the agent produces effective cooperative actions that improve network-wide traffic efficiency.
>
>
> [1] Kahneman D. Thinking, fast and slow[M]. macmillan, 2011.
>
> ---
> # W2/W7: Analysis of how LLM reasoning benefits coordination
> > W2: The cooperative reasoning process relies on textual prompting, but there is little analysis of why or how the LLM reasoning benefits coordination beyond the cached contextual hints. No ablation or visualization clarifies what the LLM is learning about spatial-temporal dependencies.
>
> > W7: ... limited insights into why the model works, missing a deeper understanding of LLM-agent cooperation dynamics.
>
> We sincerely appreciate the reviewer’s constructive feedback. To address these concerns, we provide both (i) ablation study results to verify whether the LLM truly leverages spatial–temporal contexts, and (ii) a case study illustrating how reasoning contributes to coordination across intersections.
>
>
> ## Does CoLLMLight effectively utilize spatiotemporal information?
>
> We conducted ablation studies on the NewYork dataset to examine whether the LLM relies on spatial and temporal contexts during decision-making. In each ablation, we removed specific components from the input prompts.
>
> **Table 1: Ablation Study Results (Metric: ATT, lower is better)**
>
> | Method | NewYork 1 | NewYork 2 |
> | :--- | :---: | :---: |
> | CoLLMLight (Full) | **1000.4** | **1345.1** |
> | w/o Temporal Context | 1155.1 | 1477.3 |
> | w/o Spatial Context | 1112.6 | 1443.5 |
> | w/o Both | 1168.7 | 1542.2 |
>
> Removing either temporal or spatial information causes a clear performance drop. Notably, removing spatial context results in a substantial degradation, empirically confirming that the agent actively uses neighboring-intersection information rather than operating as an isolated controller.

---

> > ### Author Response · Authors · 2025-11-24
> > **Response to Reviewer TspF Part 2**
> >
> > ## How does reasoning benefit coordination, and how does coordination emerge?
> >
> > To further illustrate how SR facilitates cooperation across intersections, we include a detailed case study in the Appendix A.7.
> >
> > A concise version is summarized below.
> >
> > **Case description.**
> > At the current timestep, NT and its upstream lanes contain a large number of vehicles, but NT’s downstream lanes are already saturated. WT also maintains a growing queue, yet its downstream remains uncongested. This configuration forms a clear spillback-risk scenario.
> >
> > **SR process:**
> >
> > 1. **Identification of critical lanes.**
> >    The SR module identifies NT and WT as critical lanes and assesses their upstream and downstream pressures.
> >
> > 2. **Reflection on previous actions.**
> >    The reasoning notes that earlier activation of ETWT helped prevent further spillback into NT’s already congested downstream lanes.
> >
> > 3. **Conditional planning based on network context.**
> >    The SR module generates a clear conditional plan:
> >    - If NT’s downstream becomes sufficiently cleared, releasing NT is feasible.
> >    - Otherwise, prioritize ETWT to avoid deteriorating the downstream bottleneck.
> >
> > **RD process:**
> >
> > Guided by SR’s conditional plan and the current observation (NT’s downstream remains congested), the RD module decides to hold NTST and activate ETWT, preventing additional congestion.
> >
> > **How coordination emerges**
> >
> > Coordination emerges when an intersection detects potential congestion risks in its neighboring lanes. In CoLLMLight, this coordination is achieved through the SR→RD pipeline: the SR module first anticipates network-level conditions and prepares a conditional plan, and the RD module subsequently executes the most suitable action based on the real-time observation.
> >
> > This mechanism is reflected clearly in the present scenario. Although NT has the largest queue, SR identifies that releasing NT would worsen its already congested downstream lanes and therefore recommends prioritizing ETWT unless downstream conditions improve. When the RD module observes that the downstream blockage remains unresolved, it follows the SR plan and activates ETWT to prevent further congestion propagation and maintain stable flow.
> >
> > This behavior fundamentally differs from independent agents, which optimize only for local queue reduction. Such agents would release NT regardless of downstream conditions, thereby amplifying congestion.
> >
> > ---
> > # W3: Clarification on real-time feasibility and deployment considerations
> > > Although the asynchronous module mitigates latency, the experiments rely on simulation rather than real-time control. It’s unclear whether the system can truly operate under realistic time constraints with communication and inference overhead among dozens of intersections
> >
> > We thank the reviewer for raising this important point.
> >
> > **Real-time feasibility is fully achievable in practice.**
> > Our communication payload is extremely lightweight: each agent transmits less than 1 KB of structured numerical lane features per signal interval. This results in minimal bandwidth requirements and is well within the capability of modern traffic-signal communication infrastructures.
> >
> > **Communication latency is also realistic.**
> > While our simulation assumes ideal communication, existing ITS deployments already satisfy such latency bounds. The FHWA feasibility study [2] reports that Dedicated Short-Range Communications (DSRC) can achieve point-to-point delays below 2 ms, demonstrating that millisecond-level coordination is feasible in real-world settings.
> >
> > **Inference time is well within operational limits.**
> > Our inference time analysis in Figure 3 shows that the RD module exhibits very low latency, with average delays of **1.02 / 1.80 / 2.46 s** when jointly processing batches of 1, 5, and 10 intersections on a 2×A800 setup. These delays are far below typical yellow-light intervals (3-6 s) [3], fully meeting real-time signal control requirements.
> >
> > Practical deployments may achieve even lower latency via quantization, model distillation, or hardware acceleration. Moreover, as LLM efficiency continues to improve, inference throughput is expected to increase accordingly [4].
> >
> > Overall, both communication and computation requirements of CoLLMLight are compatible with real-world deployment in modern traffic-signal control systems.
> >
> > [2] FHWA: DSRC Implementation Feasibility Study.
> >
> > [3] McGee Sr H, Moriarty K, Gates T J. Guidelines for timing yellow and red intervals at signalized intersections[J]. Transportation research record, 2012, 2298(1): 1-8.
> >
> > [4] Xiao C, Cai J, Zhao W, et al. *Densing Law of LLMs*. Nature Machine Intelligence, 2025.
> >
> > ---

---

> ### Author Response · Authors · 2025-11-24
> **Response to Reviewer TspF Part 3**
>
> # W4/W6: Variance and Statistical Significance Analysis and Additional Baseline Methods
>
> Thank the reviewer’s feedback.
> We have add a new section in Appendix (A.8) to report variance and statistical significance analysis. In addition, we include three new MARL baselines: CoSLight [5] (transformer-based MARL), X-Light [6] (meta-transformer MARL), and DuaLight [7] (scenario-shared and scenario-specific learning).
> Below is the results:
>
> **Table 2: Zero-shot performance comparison with variance**
> | Models | NY1 ATT | NY1 AWT | NY1 AQL | NY2 ATT | NY2 AWT | NY2 AQL | Jinan ATT | Jinan AWT | Jinan AQL | Hang ATT | Hang AWT | Hang AQL |
> |--------|---------|---------|---------|---------|---------|---------|------------|------------|------------|----------|----------|----------|
> | FixedTime | 1535.6 ± 0 | 290.1 ± 0 | 3173.6 ± 0 | 1771.7 ± 0 | 428.7 ± 0 | 4523.1 ± 0 | 441.2 ± 0 | 66.7 ± 0 | 294.1 ± 0 | 616.0 ± 0 | 74.0 ± 0 | 301.3 ± 0 |
> | MaxPressure | 1223.3 ± 0 | 153.3 ± 0 | 2410.2 ± 0 | 1566.8 ± 0 | 255.5 ± 0 | 3936.5 ± 0 | 273.2 ± 0 | 38.2 ± 0 | 106.6 ± 0 | 325.3 ± 0 | 49.6 ± 0 | 68.9 ± 0 |
> | MPLight | 1492.2 ± 42.80 | 262.2 ± 8.75 | 3054.1 ± 87.46 | 1711.0 ± 27.11 | 364.6 ± 7.43 | 4324.5 ± 62.47 | 482.2 ± 35.14 | 84.6 ± 5.60 | 340.6 ± 39.49 | 496.2 ± 21.28 | 64.7 ± 2.87 | 197.5 ± 15.16 |
> | AttendLight | 1267.7 ± 20.07 | 292.8 ± 16.46 | 2557.2 ± 22.74 | 1755.3 ± 25.52 | 496.8 ± 9.12 | 4571.8 ± 36.32 | 310.2 ± 6.24 | 63.4 ± 1.42 | 144.5 ± 5.01 | 327.2 ± 2.27 | 72.9 ± 3.66 | 69.0 ± 0.71 |
> | PressLight | 1687.2 ± 21.25 | 508.4 ± 13.92 | 3542.7 ± 48.34 | 1894.0 ± 14.96 | 489.7 ± 38.56 | 4820.7 ± 36.13 | 410.1 ± 15.32 | 139.2 ± 9.21 | 270.4 ± 18.75 | 600.0 ± 9.41 | 259.2 ± 6.28 | 301.1 ± 5.91 |
> | CoLight | 1427.9 ± 39.05 | 246.7 ± 11.44 | 2923.0 ± 71.00 | 1753.5 ± 43.98 | 457.7 ± 27.27 | 4517.6 ± 108.36 | 474.4 ± 6.82 | 90.8 ± 4.13 | 333.1 ± 5.68 | 530.2 ± 17.32 | 90.3 ± 5.22 | 228.1 ± 12.78 |
> | E-CoLight | 1266.0 ± 16.98 | 281.3 ± 11.36 | 2664.2 ± 67.19 | 1645.7 ± 20.93 | 437.3 ± 9.17 | 4327.5 ± 25.50 | 844.8 ± 99.41 | 489.9 ± 66.87 | 799.2 ± 76.91 | 874.5 ± 38.01 | 466.8 ± 57.91 | 540.8 ± 38.11 |
> | A-CoLight | 1037.9 ± 17.13 | 185.5 ± 16.75 | 1845.8 ± 93.74 | 1428.9 ± 17.68 | 359.8 ± 2.10 | 3665.6 ± 9.52 | 392.9 ± 3.43 | 144.7 ± 3.56 | 242.9 ± 3.81 | 428.0 ± 5.73 | 235.9 ± 6.38 | 156.5 ± 6.21 |
> | **CosLight** | 1851.4 ± 7.60 | 761.8 ± 12.36 | 3225.1 ± 17.77 | 1986.9 ± 15.47 | 816.3 ± 14.08 | 4025.8 ± 43.08 | 875.9 ± 4.25 | 752.5 ± 8.69 | 890.3 ± 4.77 | 848.7 ± 5.21 | 696.5 ± 15.72 | 557.6 ± 5.18 |
> | **X-Light** | 1608.7 ± 11.27 | 281.6 ± 6.01 | 3313.9 ± 25.88 | 1815.6 ± 12.78 | 368.8 ± 8.16 | 4635.9 ± 9.36 | 661.1 ± 46.84 | 117.3 ± 9.83 | 548.6 ± 57.19 | 695.8 ± 25.45 | 117.5 ± 11.46 | 352.6 ± 16.71 |
> | **DuaLight** | 1102.3 ± 14.79 | 337.3 ± 4.45 | 2026.1 ± 57.64 | 1487.7 ± 14.53 | 467.5 ± 17.84 | 3952.3 ± 54.67 | 425.5 ± 13.95 | 230.7 ± 25.77 | 289.7 ± 21.02 | 558.5 ± 39.15 | 435.1 ± 36.22 | 277.3 ± 35.26 |
> | LLMLight-8B | 1187.4 ± 12.19 | 143.1 ± 5.17 | 2297.9 ± 3.25 | 1599.4 ± 6.65 | 388.7 ± 14.44 | 4074.6 ± 10.55 | 268.6 ± 1.41 | 40.5 ± 1.06 | 102.2 ± 1.58 | 312.0 ± 0.62 | 39.5 ± 0.63 | 58.1 ± 0.54 |
> | Llama3.1-8B | 1196.5 ± 23.08 | 242.3 ± 70.69 | 2342.8 ± 46.57 | 1540.7 ± 12.76 | 388.7 ± 1.49 | 3917.7 ± 33.17 | 281.3 ± 2.06 | 44.7 ± 1.71 | 114.3 ± 1.88 | 333.8 ± 0.09 | 45.6 ± 0.18 | 73.5 ± 0.12 |
> | Qwen3-8B | 1109.2 ± 20.26 | 116.6 ± 3.33 | 2134.1 ± 45.96 | 1435.9 ± 6.48 | 245.0 ± 6.06 | 3666.7 ± 34.55 | 268.8 ± 0.30 | 37.3 ± 0.04 | 100.8 ± 0.10 | 311.1 ± 2.89 | 40.3 ± 0.04 | 56.5 ± 1.33 |
> | R1-8B | 1253.9 ± 1.25 | 189.1 ± 16.38 | 2455.4 ± 59.03 | 1551.7 ± 8.30 | 321.2 ± 6.06 | 3910.7 ± 9.97 | 329.8 ± 0.37 | 76.9 ± 0.66 | 167.9 ± 0.67 | 372.8 ± 1.53 | 91.5 ± 0.09 | 105.8 ± 2.91 |
> | Qwen3-32B | 1040.8 ± 10.59 | 114.6 ± 2.37 | 1939.4 ± 27.05 | 1381.2 ± 3.10 | 237.5 ± 4.72 | 3510.3 ± 32.01 | 277.1 ± 0.33 | 38.5 ± 0.10 | 110.6 ± 0.54 | 314.8 ± 1.25 | 40.9 ± 0.04 | 59.5 ± 1.58 |
> | R1-32B | 1117.2 ± 5.45 | 159.3 ± 11.63 | 2050.2 ± 34.41 | 1375.4 ± 6.21 | 246.3 ± 4.38 | 3463.1 ± 54.24 | 272.6 ± 0.30 | 37.7 ± 0.58 | 105.3 ± 0.23 | 310.9 ± 1.65 | 37.1 ± 0.04 | 56.3 ± 1.45 |
> | Llama3.1-70B | 1082.6 ± 4.33 | 101.3 ± 9.00 | 2060.5 ± 12.99 | 1427.6 ± 8.37 | 245.0 ± 4.79 | 3644.8 ± 21.06 | 269.2 ± 0.28 | 36.1 ± 0.11 | 101.9 ± 0.15 | 313.7 ± 2.12 | 34.8 ± 0.02 | 58.5 ± 1.59 |
> | **CoLLMLight-8B** | **1000.4 ± 8.67** | **90.5 ± 3.66** | **1816.8 ± 28.83** | **1345.1 ± 5.13** | **167.6 ± 4.96** | **3394.2 ± 46.24** | **267.9 ± 0.37** | **34.8 ± 0.37** | **100.4 ± 0.68** | **308.5 ± 1.12** | **33.5 ± 0.02** | **54.9 ± 0.95** |

---

> ### Author Response · Authors · 2025-11-24
> **Response to Reviewer TspF Part 4**
>
> In our main experiment, all learning-based controllers are trained on the SynTrain scenario and evaluated in a zero-shot manner on unseen cities. Although MARL approaches such as DuaLight achieve strong results on the training road network, they exhibit substantial performance degradation when transferred to new networks.
>
> In contrast, CoLLMLight consistently demonstrates the best generalization performance, benefiting from its semantic-based reasoning capability and our cooperation optimization strategy.
>
> ## Statistical significance analysis
> Each model is evaluated using five random seeds (11, 42, 1024, 2025, 20). We compute variance and conduct t-tests comparing all baselines against CoLLMLight-8B.
>
> Across almost all datasets and metrics, CoLLMLight-8B shows highly significant improvements with p values below 0.001. Only a few isolated cases fall outside this range. Non-significant differences (p greater or equal to 0.05) appear for A-CoLight on New York 1 AQL, Qwen3-8B on Jinan AQL, and R1-32B on Hangzhou AQL. Moderately significant results (0.01 to 0.05) occur for R1-32B on New York 2 AQL and Llama3.1-70B on New York 1 AWT. One intermediate significance case (0.001 to 0.01) is observed for R1-32B on Hangzhou ATT.
>
> Except for these rare exceptions, all remaining comparisons demonstrate that CoLLMLight-8B consistently and significantly outperforms all alternative models.
>
> [5] Ruan J. et al. CoSLight. KDD 2024.
>
> [6] Jiang H. et al. X-Light. IJCAI 2024.
>
> [7] Lu J. et al. DuaLight. AAMAS 2024.
>
> ---
> # W5: Human evaluation
> We thank the reviewer for raising this concern.
>
> To assess whether CoLLMLight’s decisions are interpretable and safe, we conducted an expert inspection study involving four transportation researchers familiar with traffic-signal control. The evaluation is based on 100 sampled outputs (including both SR and RD results from CoLLMLight-8B), and each item is scored on a 1–5 scale across five dimensions: interpretability, internal consistency, decision soundness, reasoning validity, and risk-awareness. The results are as follows:
>
> **Table 3. Human evaluation results**
> | Metric | Mean ± Std |
> |--------|------------|
> | Interpretability | 4.78 ± 0.17 |
> | Consistency | 4.59 ± 0.26 |
> | Decision soundness | 4.71 ± 0.12 |
> | Reasoning validity | 4.74 ± 0.14 |
> | Risk-awareness | 4.83 ± 0.11 |
> | Overall | 4.73 ± 0.13 |
>
> Experts consistently observed that CoLLMLight produces clear and logically coherent reasoning traces, accurately identifies potential safety risks (such as downstream blockage), and selects signal phases aligned with established traffic-engineering principles.
>
> ---
> # Q1: How are LLM reasoning traces (SR) concretely represented and reused during decision making?
> The SR reasoning traces are reused in natural-language form as part of the contextual input to the real-time decision (RD) module, enabling it to quickly grasp the broader traffic context, recognize cooperation opportunities, and make an appropriate signal decision based on real-time observations.
>
> ---
>
> # Q2: How is the asynchronous communication among intersections implemented? Do agents share text tokens, numeric signals, or both?
> At each timestep, intersections exchange structured numerical traffic-state data with their collaborating neighbors (e.g., queue lengths, waiting times, lane occupancy), not text.
> All raw data remain in numerical form during communication to keep the bandwidth minimal.
>
> Once received, each local agent converts these numerical features into a textual description locally, which is then used as part of the input for LLM-based reasoning.
>
> ---
> # Q3: How robust is CoLLMLight to communication failures or stale reasoning caches in dynamic conditions?
>
> To answer this question, we introduced new experiments on two scenarios:
>
> 1. **Stale SR:** SR guidance becomes outdated by one timestep with 50% probability.
> 2. **Communication failure:** one neighbor’s state is randomly hidden with 50% probability.
>
> **Table 4. Robustness test of CoLLMLight (ATT, lower is better)**
> | Scenario / Dataset            | New York 1 | New York 2 |
> |-------------------------------|------------|------------|
> | **Normal**                    | **1000.4** | **1345.1** |
> | **Stale SR**                  | 1017.7     | 1380.4     |
> | **Communication failure**     | 1028.1     | 1381.7     |
>
> The performance degradation remains small, showing that the asynchronous SR–RD architecture is robust.
> This is because SR relies on historical information for its reasoning, which helps reduce the impact of partial communication failures, while the RD module adjusts based on real-time observations rather than blindly following outdated SR outputs.
>
> ---
> # CLosing note
> We deeply appreciate the reviewer’s valuable comments once again.
> We hope our response adequately address all concerns and will be viewed favorably in the final evaluation.
> If there are any remaining issues or questions, we would be happy to elaborate further.

---

### Official Review · Reviewer_9P3T · 2025-10-31

**Soundness:** 3
**Presentation:** 3
**Contribution:** 3
**Rating:** 8
**Confidence:** 3

**Summary:**

The paper addresses a relevant and emerging topic in large-scale urban traffic signal control by extending LLM-based single-intersection agents to a cooperative multi-agent framework.

The proposed CoLLMLight architecture integrates asynchronous reasoning–decision decoupling with cost-aware reinforcement optimization, aiming to balance coordination performance and reasoning efficiency.

The idea is interesting and experimentally validated across multiple city networks.

However, several aspects—including the theoretical motivation, reward design rigor, and consistency of coordination—require further clarification and analysis to fully establish the framework’s soundness and generality.

**Strengths:**

1.Explores a novel and timely research direction by bridging large language models (LLMs) with multi-agent coordination for network-wide traffic signal control.

2.The proposed Spatiotemporal Reasoning–Real-Time Decision (SR–RD) decoupling effectively mitigates LLM latency issues in real-time control.

3.Experiments on multiple city-scale networks demonstrate the scalability of the approach beyond toy environments.

4.The paper is generally well-written and clearly organized, making it accessible to both MARL and LLM audiences.

**Weaknesses:**

1.The paper does not clearly explain why large language models (LLMs) are more suitable for multi-interaction cooperation than conventional multi-agent reinforcement learning (MARL) frameworks.

2.There are fewer recent baselines in the comparative experiments.

3.The SR (Spatiotemporal Reasoning) and RD (Real-Time Decision) modules operate asynchronously, and the reasoning latency of SR is not fixed. This could lead to desynchronized decisions among intersections.

4.Cooperation among intersections is primarily achieved through prompt aggregation and message sharing at the text level. The framework would benefit from a more explicit or interpretable mechanism for cooperative policy modeling to reveal how coordination behaviors emerge across agents.

**Questions:**

1.The paper does not clearly explain why large language models (LLMs) are more suitable for multi-interaction cooperation than conventional multi-agent reinforcement learning (MARL) frameworks. A theoretical or conceptual discussion on the limitations of existing RL-based coordination strategies is necessary to justify the shift toward LLM-based control.


2. Comparing more recent baselines will enhance the persuasiveness of the experimental results.


3. The paper should analyze how asynchronous reasoning affects network-wide coordination and feedback consistency.

---

> ### Author Response · Authors · 2025-11-23
> **Response to Reviewer 9P3T Part 1**
>
> We sincerely appreciate the reviewer’s thoughtful feedback and are encouraged by the positive remarks on the novelty and contributions of our work.
> To address the concerns raised, we provide detailed responses below.
>
> ---
> # W1/Q1: Discussion of CoLLMLight and conventional MARL-based methods
> > W1: The paper does not clearly explain why LLMs are more suitable for multi-interaction cooperation than conventional multi-agent reinforcement learning (MARL) frameworks.
>
> > Q1: A theoretical or conceptual discussion on the limitations of existing RL-based coordination strategies is necessary to justify the shift toward LLM-based control.
>
> We thank the reviewer for raising this important point. In the revised paper, we have expanded the discussion to clarify this methodological shift:
>
> **Limitations of Conventional MARL**
> Traditional MARL approaches encode traffic states into numerical latent vectors and model inter-intersection relations using architectures such as transformers or GATs. While effective on the training topology, their cooperation remains *implicit*—embedded within black-box neural networks. As a result, these policies often struggle to generalize to unseen road networks without retraining, suffering from roadnet mismatch and overfitting.
>
> **The CoLLMLight Advantage**
> CoLLMLight retains the multi-agent structure but shifts from vector-based state encoding to structured textual representations, enabling the model to perform explicit semantic reasoning (e.g., “release NT lane to prevent upstream spillback”). This design introduces two advantages:
> * (i) Improved interpretability of the cooperative behavior,
> * (ii) Topology-agnostic generalization, allowing zero-shot deployment on unseen networks.
>
> To support this shift, our framework incorporates two key components:
> * An Asynchronous Cooperative Decision Architecture, which separates heavy reasoning from real-time signal execution to maintain responsiveness.
> * A Cost-Aware Cooperation Optimization Strategy, combined with RL, which improves reasoning efficiency and grounds the semantic reasoning process in environment feedback to ensure valid and coherent strategies.
>
> ---
> # W2/Q2 Experiments on more baselines
> We thank the reviewer for the suggestion.
> To further strengthen the empirical comparison, we have added three recent and representative MARL baselines: **CoSLight** [1] (collaborator selection), **X-Light** [2] (meta-Transformer MARL), and **DuaLight** [3] (scenario-shared + scenario-specific knowledge).
> We trained these methods on the **Syn-Train** dataset and evaluated their **zero-shot transfer performance** on four unseen real-world datasets.
> The updated results are presented below:
>
> **Table 1: Zero-shot performance (ATT/AWT/AQL) comparison across different datasets**
> | Model              | New York 1              | New York 2              | Jinan                   | Hangzhou               |
> |--------------------|--------------------------|--------------------------|--------------------------|--------------------------|
> | FixedTime          | 1535.6/290.1/3173.6     | 1771.7/428.7/4523.1     | 441.2/66.7/294.1        | 616.0/74.0/301.3         |
> | MaxPressure        | 1223.3/153.3/2410.2     | 1566.8/255.5/3936.5     | 273.2/38.2/106.6        | 325.3/49.6/68.9          |
> | MPLight            | 1492.2/262.2/3054.1     | 1711.0/364.6/4324.5     | 482.2/84.6/340.6        | 496.2/64.7/197.5         |
> | AttendLight        | 1267.7/292.8/2557.2     | 1755.3/496.8/4571.8     | 310.2/63.4/144.5        | 327.2/72.9/69.0          |
> | PressLight         | 1687.2/508.4/3542.7     | 1894.0/489.7/4820.7     | 410.1/139.2/270.4       | 600.0/259.2/301.1        |
> | CoLight            | 1427.9/246.7/2923.0     | 1753.5/457.7/4517.6     | 474.4/90.8/333.1        | 530.2/90.3/228.1         |
> | E-CoLight          | 1266.0/281.3/2664.2     | 1645.7/437.3/4327.5     | 844.8/489.9/799.2       | 874.5/466.8/540.8        |
> | A-CoLight          | 1037.9/185.5/1845.8     | 1428.9/359.8/3665.6     | 392.9/144.7/242.9       | 428.0/235.9/156.5        |
> | **CoSLight**       | 1851.4/761.8/3225.1     | 1986.9/816.3/4025.8     | 875.9/752.5/890.3       | 848.7/696.5/557.6        |
> | **X-Light**        | 1608.7/281.6/3313.9     | 1815.6/368.8/4635.9     | 661.1/117.3/548.6       | 695.8/117.5/352.6        |
> | **DuaLight**       | 1102.3/337.3/2026.1     | 1487.7/467.5/3952.3     | 425.5/230.7/289.7       | 558.5/435.1/277.3        |
> | LLMLight-8B        | 1187.4/143.1/2297.9     | 1599.4/388.7/4074.6     | 268.6/40.5/102.2        | 312.0/39.5/58.1          |
> | **CoLLMLight-8B**  | **1000.4/90.5/1816.8**  | **1345.1/167.6/3394.2** | **267.9/34.8/100.4**    | **308.5/33.5/54.9**      |

---

> ### Author Response · Authors · 2025-11-23
> **Response to Reviewer 9P3T Part 2**
>
> **Table 2: Comparative Performance of Learning-based Methods on Syn-Train**
> | Method | ATT | AWT | AQL |
> | :--- | :--- | :--- | :--- |
> | MPLight | 979.03 | 504.66 | 568.29 |
> | AttendLight | 753.09 | 152.85 | 502.38 |
> | PressLight | 715.45 | 403.20 | **371.90** |
> | CoLight | 826.44 | 260.96 | 492.87 |
> | E-CoLight | 716.40 | 361.47 | 521.49 |
> | A-CoLight | 671.60 | 348.82 | 491.97 |
> | **CoSLight** | 949.97 | 646.26 | 502.24 |
> | **X-Light** | 989.18 | 183.25 | 525.46 |
> | **DuaLight** | **621.63** | 531.31 | 451.08 |
> | LLMLight-8B | 1256.93 | 411.15 | 790.06 |
> | **CoLLMLight-8B** | 868.81 | **110.02** | 579.41 |
>
> Comparing the zero-shot (Table 1) and supervised (Table 2) results reveals a critical insight: while MARL methods like **DuaLight** achieve superior fitting performance on the training set, they suffer from severe performance degradation in zero-shot scenarios. This stark contrast confirms that conventional MARL approaches tend to **overfit specific topologies** and struggle to transfer learned cooperation behavior to unseen networks.
> Conversely, CoLLMLight emerges as the strongest generalizer across all unseen datasets.
> This validates that our LLM-based approach captures universal traffic control logic via semantic reasoning rather than merely memorizing topological patterns.
>
> [1] Ruan et al., CosLight, KDD’24.
> [2] Jiang et al., X-Light, IJCAI’24.
> [3] Lu et al., DuaLight, AAMAS’24.
>
> ---
>  # W3/Q3: On the effect of asynchronous reasoning on network-wide decision consistency
>  > W3: The SR (Spatiotemporal Reasoning) and RD (Real-Time Decision) modules operate asynchronously, and the reasoning latency of SR is not fixed. This could lead to desynchronized decisions among intersections.
>
>  > Q3: The paper should analyze how asynchronous reasoning affects network-wide coordination and feedback consistency.
>
> **Role of SR and RD.**
> In our implementation, the SR module performs spatiotemporal reasoning (over historical signals, traffic evolution, and spatial relations) and produces a cooperative guidance plan, which is cached as context.
> The RD module makes real-time decisions at every signal interval based on **(i)** the latest traffic observations (local and neighbor states) and **(ii)** the most recent SR guidance.
> Importantly, RD also evaluates whether the cached SR guidance remains suitable for the current traffic condition. If the guidance is outdated, RD relies primarily on real-time observations to avoid misaligned or conflicting decisions.
> This design ensures that intersections remain synchronized with the current traffic state, even when SR completion is delayed.
>
> ** Robustness Test**
> To test robustness under network instability, we further introduced additional experiments on two scenarios:
> 1. **Stale SR:** SR guidance becomes outdated by one timestep with 50% probability.
> 2. **Communication failure:** one neighbor’s state is randomly hidden with 50% probability.
>
> **Table 3. Robustness test of CoLLMLight (ATT, lower is better)**
>
> | Scenario / Dataset            | New York 1 | New York 2 |
> |-------------------------------|------------|------------|
> | **Normal**                    | **1000.4** | **1345.1** |
> | **Stale SR**                  | 1017.7     | 1380.4     |
> | **Communication failure**     | 1028.1     | 1381.7     |
>
> The performance degradation remains small, showing that the asynchronous SR–RD architecture is robust.
> This is because SR relies on historical information for its reasoning, which helps reduce the impact of communication failures, while the RD module adjusts based on real-time observations rather than blindly following outdated SR outputs.

---

> ### Author Response · Authors · 2025-11-23
> **Response to Reviewer 9P3T Part 3**
>
> # W4: Clarification of the Explicit and Interpretable Cooperation Mechanism
>  > Cooperation among intersections is primarily achieved through prompt aggregation and message sharing at the text level. The framework would benefit from a more explicit or interpretable mechanism for cooperative policy modeling to reveal how coordination behaviors emerge across agents.
>
> We appreciate the reviewer's insightful suggestion.
>
> In our current framework, each agent is assigned a set of neighboring intersections as its cooperation scope, together with structured spatiotemporal information from these neighbors. The LLM-based policy then performs semantic reasoning over this context to determine *how* to coordinate with specific neighbors to improve network-level efficiency.
>
> This cooperation is already  explicit: the reasoning traces reveal *which neighbor* the agent aims to coordinate with and *why* (e.g., “prioritize the NT–ST corridor to prevent spillback from the northern intersection”). As shown in our case studies (Appendix A.7), these explanations align well with the resulting control actions, offering a degree of interpretability that conventional MARL policies typically lack.
>
> The reviewer’s suggestion—developing a more explicit cooperative modeling mechanism, such as allowing agents to dynamically select collaborators or exchange richer cooperative messages—would further enhance the flexibility and expressiveness of inter-agent coordination. We consider this a promising direction and plan to explore it in future work.
>
> ---
> # CLosing note
> We thank the reviewer once again for the valuable feedback.
> We hope that the clarifications and additional experiments satisfactorily address all concerns.
> We would be glad to provide any further details if needed.

---

### Official Review · Reviewer_Vb9x · 2025-10-31

**Soundness:** 2
**Presentation:** 2
**Contribution:** 3
**Rating:** 4
**Confidence:** 4

**Summary:**

This paper introduces CoLLMLight, the first framework that uses cooperative large language model agents for managing traffic signals across entire road networks. Unlike existing LLM-based controllers that operate intersections independently, CoLLMLight enables intersections to share spatiotemporal information and coordinate decisions asynchronously, improving network-wide traffic flow. The model includes a cost-aware cooperation optimization mechanism that balances reasoning depth and computation efficiency through adaptive reasoning and reinforcement learning. Experiments on four real-world traffic networks show that CoLLMLight outperforms traditional, reinforcement learning, and prior LLM-based methods in all evaluation metrics, while maintaining real-time responsiveness.

**Strengths:**

1. The problem discussed in the paper is a very important and interesting problem.

2. The ablation study of the work is comprehensive enough to understand how different parts of the framework contribute, especially in the inference time part.

**Weaknesses:**

1. Some important details are missing in the paper. For example, what is the base model that the authors try to optimize.  And the introduction to the framework is somehow vague, lacking a case study to show how the system really works.

2. The experimental setting is not very standard for RL-based baselines. Normally, in the original paper, RL-based methods are trained on the same map as the evaluation setup, with a different traffic scenario. However, this paper uses a transfer learning style evaluation. Though I understand LLM-based methods have a better generalization on this transfer style evaluation, which is also one of the advantages of the LLM-based method,  I hope to see a comparison for a normal setting for the RL-based method to better understand the performance.

3. This paper lacks solid proof of the effectiveness of the proposed framework. In Table3, the results show that the proposed method will perform worse than LLMLight-8B (From Table 1) without any policy refinement. Therefore, I doubt that the performance gain is coming from the training process in policy refinement instead of the multi-agent framework.

**Questions:**

Please see the weakness part

---

> ### Author Response · Authors · 2025-11-23
> **Response to Reviewer Vb9x Part 1**
>
> We sincerely thank the reviewer for the detailed and constructive feedback.
> We are grateful for the recognition of the importance of our problem and the value of our ablation study.
> Below, we provide point-by-point responses addressing all concerns, and we are happy to clarify any remaining aspects if needed.
>
> ---
> # W1: Clarification of Methodological Details
> > Some important details are missing in the paper. For example, what is the base model that the authors try to optimize. And the introduction to the framework is somehow vague, lacking a case study to show how the system really works.
>
> Thank you for pointing this out.
> As stated in Appendix A.5 (Implementation Details), CoLLMLight-8B is obtained by fine-tuning a Llama-3.1-8B base model.
> We have revised the paper accordingly and now explicitly clarify this in the experimental section.
>
> To provide a clearer overview of how the system operates, we include a structured procedural description below.
> We also refer the reviewer to the case study in Appendix A.7, which compares CoLLMLight’s decision process with that of the base model.
>
> ### **Procedure of CoLLMLight at time step $t$**
>
> **Input:**
> - Latest traffic state of intersection $i$ and its collaborators at time $t$: $O_i^t$
> - Historical signals and traffic states of $i$ and its collaborators: $H_i^{t-\Delta t:t-1}$
> - Spatiotemporal-aware cooperative reasoning (SR) results from previous step (optional): $R_i^{t-1}$
>
> **Output:**
> - Selected signal phase at time $t$: $a_i^t$
> - New SR results: $R_i^t$
>
> ---
>
> 1. At time step $t$, start two asynchronous processes:
>    - real-time decision (**RD**) for selecting the signal at the current timestep
>    - spatiotemporal-aware cooperative reasoning (**SR**) for generating cooperative guidance that will support RD decisions in future timesteps
>
> 2. **Real-time Decision (RD) at time $t$:**
>    2.1 Construct a decision prompt $D_i^t$ using:
>        - current observation $O_i^t$
>        - previous cached SR results $R_i^{t-1}$
>    2.2 Query the RD module with $D_i^t$ to obtain action $a_i^t$.
>    2.3 Execute the selected signal phase $a_i^t$ at intersection $i$.
>
> 3. **Spatiotemporal-aware Cooperative Reasoning (SR) at time $t$:**
>    3.1 Construct a reasoning context $C_i^t$ using:
>        - historical information $H_i^{t-\delta t:t-1}$
>        - current observation $O_i^t$
>    3.2 Feed $C_i^t$ to the SR module.
>    3.3 Adaptively select and execute the necessary SR reasoning steps, such as:
>        - critical lane identification
>        - spatial interaction analysis (upstream and downstream)
>        - temporal pattern analysis
>        - reflection on past signal decisions
>        *(Skipping unnecessary steps in simple or uncongested traffic situations.)*
>    3.4 Produce SR results $R_i^t$.
>
> ---

---

> ### Author Response · Authors · 2025-11-23
> **Response to Reviewer Vb9x Part 2**
>
> # W2: Comparison with RL-based methods under a normal setting
> > ... I hope to see a comparison for a normal setting for the RL-based method to better understand the performance.
>
> We appreciate the reviewer’s insightful comment regarding the experimental setting.
>
> Our primary experiments adopt a transfer-style setting, where all methods are trained on the same training roadnet but evaluated on **unseen roadnets and traffic scenarios**. This design measures generalization performance, which aligns with the practical deployment goal of LLM-based controllers—zero-shot application to new cities without retraining.
>
> To address the reviewer’s request for a standard same-map comparison, we report in the appendix (included in the initial submission) the results on the Syn-Train (4×4 grid) network, where **each learning-based method is trained and evaluated on the same roadnet**. We also provide an expanded analysis in the revised version. The results are reproduced below:
>
> **Table: Performance of learning-based methods on Syn-Train**
>
> | Method        | ATT     | AWT     | AQL     |
> |---------------|---------|---------|---------|
> | MPLight       | 979.03  | 504.66  | 568.29  |
> | AttendLight   | 753.09  | 152.85  | 502.38  |
> | PressLight    | 715.45  | 403.20  | **371.90** |
> | CosLight      | 949.97  | 646.26  | 502.24  |
> | X-Light       | 989.18  | 183.25  | 525.46  |
> | DuaLight      | **621.63** | 531.31 | 451.08 |
> | CoLight       | 826.44  | 260.96  | 492.87  |
> | E-CoLight     | 716.40  | 361.47  | 521.49  |
> | A-CoLight     | **671.60** | 348.82 | 491.97 |
> | LLMLight      | 1256.93 | 411.15  | 790.06  |
> | CoLLMLight    | 868.81  | **110.02** | 579.41 |
>
> As shown, RL-based methods—particularly A-CoLight, DuaLight, and PressLight—perform strongly in this setting because they are fully trained on this specific map.
>
> Compared with conventional RL approaches, CoLLMLight employs an LLM-based agent driven by language reasoning and optimized with our proposed strategy, and achieves competitive performance, particularly in terms of average waiting time.
>
> Moreover, by leveraging the reasoning capability of LLMs, our method demonstrates stronger generalization on unseen scenarios in the main experiments.
>
>
> ---
> # W3: Evidence supporting the effectiveness of the proposed framework
> > This paper lacks solid proof of the effectiveness of the proposed framework. In Table3, the results show that the proposed method will perform worse than LLMLight-8B (From Table 1) without any policy refinement. Therefore, I doubt that the performance gain is coming from the training process in policy refinement instead of the multi-agent framework.
>
> Thank you for this feedback. We first clarify that the LLMLight-8B reported in Table 1 is obtained by taking Llama-3.1-8B as the base model and optimizing it using the LLMLight training procedure. To directly address the reviewer’s concern, we provide below a controlled comparison between LLMLight and CoLLMLight under the **same base model** and after their respective optimization steps.
>
> **Table: Comparison of base models and optimized models on the NewYork1 dataset (ATT, lower is better)**
>
> | Method       | Base model (Llama-3.1-8B) | Optimized model |
> |--------------|---------------------------|-----------------|
> | LLMLight     | 1289.2                    | 1187.4          |
> | CoLLMLight   | 1196.5 (↓7.2%)            | 1000.4 (↓15.7%) |
>
> This comparison shows that, given the same base model, CoLLMLight achieves lower ATT both before and after optimization, indicating that the advantages of our framework are not solely attributed to the policy refinement process.
>
> Unlike LLMLight’s single-intersection decision process, CoLLMLight incorporates historical information and upstream/downstream spatial context from neighboring intersections during spatiotemporal-aware cooperative reasoning. This allows the model to consider cross-intersection interactions that single-agent methods cannot capture.
>
> At the same time, the base model alone is not capable of fully reasoning over the longer spatiotemporal context aggregated by our framework. Therefore, the cost-aware cooperation optimization is necessary: it improves cooperative decision effectiveness via environment feedback, while also encouraging shorter and more efficient reasoning.
>
> ---
> # CLosing note
> We sincerely appreciate the reviewer’s thoughtful comments once again.
> We hope that the clarifications, analyses, and additional experimental results sufficiently address all concerns and will be viewed favorably in the final evaluation.
> We are glad to provide further clarification if any additional questions arise.

---

> > ### Comment · Reviewer_Vb9x · 2025-11-27
> >
> > Thanks for the author's reply. I read it carefully and found the new experiments are very helpful. Therefore, I would like to increase my score accordingly.

---

> > > ### Author Response · Authors · 2025-11-28
> > >
> > > Thank you very much for your careful reading and encouraging feedback, which truly helped us improve the paper. We are glad that the additional clarifications and analyses in our rebuttal successfully addressed your earlier concerns.

---

### Official Review · Reviewer_jThr · 2025-10-31

**Soundness:** 3
**Presentation:** 3
**Contribution:** 3
**Rating:** 6
**Confidence:** 4

**Summary:**

The paper proposes CoLLMLight, a cooperative large language model framework for network-wide traffic signal control. Unlike existing LLM agents that manage intersections independently, CoLLMLight enables inter-intersection cooperation through an asynchronous decision architecture. It performs spatiotemporal reasoning to guide real-time signal decisions, ensuring both cooperation and responsiveness. A cost-aware optimization strategy further balances reasoning depth and efficiency using adaptive reasoning and reinforcement learning. Experiments on four real-world traffic networks show that CoLLMLight significantly outperforms existing rule-based, RL-based, and LLM-based methods, achieving better traffic flow and faster decision-making.

**Strengths:**

1. Novel Contribution: This paper proposes the first cooperative LLM-based framework for network-wide traffic signal control. It shows a new perspective for the TSC community.
2. Innovative Architecture: The asynchronous cooperative decision architecture cleverly decouples reasoning from real-time control, ensuring both deep cooperation reasoning and real-time responsiveness.
3. Adaptive and Efficient Reasoning: The introduction of cost-aware cooperation optimization and adaptive reasoning chain optimization enables the model to balance reasoning depth with computational cost, improving both efficiency and scalability.

**Weaknesses:**

1. Clarification issue: The paper should clearly explain how Cooperative Reasoning and Reflection are implemented. How the implementation fits the need of Cooperation and Reflection function, is not described.

2. The neighborhood radius (e.g., one-hop) and the precise set of lanes included are not fully specified; bandwidth, latency and the practical deployment cost, especially across large city-scale networks with many intersections, are not quantified or discussed.

**Questions:**

1. Can you provide clearer methodological details or examples/intuitions on how Cooperative Reasoning and Reflection are implemented within the pipeline?

2. Can you discuss and, if possible, test larger networks?

---

> ### Author Response · Authors · 2025-11-23
> **Response to Reviewer jThr Part 1**
>
> We deeply appreciate the reviewer's careful reading and the valuable feedback provided.
> We are encouraged by the reviewer’s acknowledgement of key strengths in our framework.
> Our detailed responses to all points follow below.
>
> ---
> # W1/Q1: Clarification of Methodological Details
> > W1: Clarification issue: The paper should clearly explain how Cooperative Reasoning and Reflection are implemented. How the implementation fits the need of Cooperation and Reflection function, is not described.
>
> > Q1: Can you provide clearer methodological details or examples/intuitions on how Cooperative Reasoning and Reflection are implemented within the pipeline?
>
> ### **The role of Cooperative Reasoning and Reflection**
>
> Cooperative Reasoning performs a comprehensive analysis of the current spatiotemporal information, including historical signals and the traffic states of the agent and its collaborators.  **Reflection is a specific reasoning step within Cooperative Reasoning.**
>
>
> We define several structured reasoning stages for Cooperative Reasoning, such as critical lane identification to prioritize high-impact movements, spatial interaction analysis to capture upstream and downstream influences, temporal pattern analysis to anticipate future congestion, and reflection on past signal decisions to avoid repeated inefficiencies.
>
> After applying our cost-aware cooperation optimization, Cooperative Reasoning produces concise yet effective outputs, and may skip unnecessary reasoning stages when operating in uncongested traffic scenarios.
>
> ### **Procedure of CoLLMLight at time step $t$**
>
> **Input:**
> - Latest traffic state of intersection $i$ and its collaborators at time $t$: $O_i^t$
> - Historical signals and traffic states of $i$ and its collaborators: $H_i^{t-\delta t:t-1}$
> - Spatiotemporal-aware cooperative reasoning (SR) results from previous step (optional): $R_i^{t-1}$
>
> **Output:**
> - Selected signal phase at time $t$: $a_i^t$
> - New SR results: $R_i^t$
>
> ---
>
> 1. At time step $t$, start two asynchronous processes:
>    - real-time decision (**RD**) for selecting the signal at the current timestep
>    - spatiotemporal-aware cooperative reasoning (**SR**) for generating cooperative guidance that will support RD decisions in future timesteps
>
> 2. **Real-time Decision (RD) at time $t$:**
>    2.1 Construct a decision prompt $D_i^t$ using:
>        - current observation $O_i^t$
>        - previous SR results $R_i^{t-1}$ (if available)
>    2.2 Query the RD module with $D_i^t$ to obtain action $a_i^t$.
>    2.3 Execute the selected signal phase $a_i^t$ at intersection $i$.
>
> 3. **Spatiotemporal-aware Cooperative Reasoning (SR) at time $t$:**
>    3.1 Construct a reasoning context $C_i^t$ using:
>        - historical information $H_i^{t-\delta t:t-1}$
>        - current observation $O_i^t$
>    3.2 Feed $C_i^t$ to the SR module.
>    3.3 Adaptively select and execute the necessary SR reasoning steps, such as:
>        - critical lane identification
>        - spatial interaction analysis (upstream and downstream)
>        - temporal pattern analysis
>        - reflection on past signal decisions
>        *(Skipping unnecessary steps in simple or uncongested traffic situations.)*
>    3.4 Produce SR results $R_i^t$.
>
> ---
> # W2: On neighborhood radius, lanes, and deployment considerations
> > The neighborhood radius (e.g., one-hop) and the precise set of lanes included are not fully specified; bandwidth, latency and the practical deployment cost, especially across large city-scale networks with many intersections, are not quantified or discussed.
>
> ### **Neighborhood radius and lane set specification**
> In our problem setting, the *neighborhood radius* is defined based on **signal-controlled lanes** rather than pure topological one-hop distance.
> In real-world urban networks, some adjacent intersections include **uncontrolled right-turn lanes**, which do not constitute a valid upstream or downstream dependency for signal coordination
> In such cases, the dependency extends through the uncontrolled lane to the next signal-controlled lane located at the corresponding two-hop intersection.
>
> As a result, the neighborhood of an agent includes:
>
> * **all directly adjacent (one-hop) intersections**, and
> * **a small subset of two-hop intersections** that become reachable when a neighboring connection passes through an uncontrolled lane.
>
> For example, in a simple (3 \times 3) grid network, if the target intersection lies at the center, its communication neighborhood includes **all 8 surrounding intersections**, as all are reachable via controlled or extended-through lanes.
>
> The **lane set for observation** follows the same dependency logic:
>
> * all controlled lanes of the target intersection, and
> * lanes from neighbor intersections that hold a *direct upstream or downstream relation* to the target intersection’s approach or exit lanes, including those reached by extending across an uncontrolled right-turn connector.

---

> ### Author Response · Authors · 2025-11-23
> **Response to Reviewer jThr Part 2**
>
> ### **Bandwidth and latency**
> Our cooperative messages are highly compact: each agent transmits **<1 KB** of lane features per signal interval, resulting in very low bandwidth usage.
>
> For latency, although our simulation assumes ideal communication, real-world ITS systems already meet this requirement. The FHWA feasibility study [1] reports that **DSRC (Dedicated Short-Range Communications) can achieve <2 ms point-to-point latency**.
>
> To test robustness under network instability, we further introduced additional experiments on two scenarios:
> 1. **Stale SR:** SR guidance becomes outdated by one timestep with 50% probability.
> 2. **Communication failure:** one neighbor’s state is randomly hidden with 50% probability.
>
> **Table 1. Robustness test of CoLLMLight (ATT, lower is better)**
>
> | Scenario / Dataset            | New York 1 | New York 2 |
> |-------------------------------|------------|------------|
> | **Normal**                    | **1000.4** | **1345.1** |
> | **Stale SR**                  | 1017.7     | 1380.4     |
> | **Communication failure**     | 1028.1     | 1381.7     |
>
> The performance degradation remains small, showing that the asynchronous SR–RD architecture is robust.
> This is because SR relies on historical information for its reasoning, which helps reduce the impact of communication failures, while the RD module adjusts based on real-time observations rather than blindly following outdated SR outputs.
>
> ### **Deployment cost**
>
> In deployment, we can collaborate with an LLM provider to serve the model via API. Current reference pricing for 8B-scale models is inexpensive (e.g., DeepInfra lists **$0.02 per 1M input tokens** and **$0.03 per 1M output tokens**). Using our measured token usage on the New York dataset (SR: 2331.69/484.20 tokens; RD: 1403.95/109.69 tokens; 2880 calls per module per day), the resulting daily cost is only **~$0.27 per intersection**.
>
> API inference costs have been steadily decreasing, suggesting that long-term operational cost will continue to drop as models become more efficient **[2]**.
>
> [1] Rayamajhi A, Balse A, Leslie E, et al. Feasibility Study and Assessment of Communications Approaches for Real-Time Traffic Signal Applications. U.S. Department of Transportation, Federal Highway Administration, 2020.
>
> [2] Xiao C, Cai J, Zhao W, et al. Densing Law of LLMs. Nature Machine Intelligence, 2025: 1–11.
>
> ---
> # Q2: Discussion on larger networks
> > Can you discuss and, if possible, test larger networks?
>
> CoLLMLight’s communication and reasoning overhead scales linearly with the number of upstream/downstream collaborators, rather than with the size of the full network.
>
> To address the reviewer’s concern, we further evaluate CoLLMLight on a **large Manhattan road network** consisting of **1176 intersections** with **46,541 vehicles per hour**.
> We follow our main experiment protocol: all learning-based methods are trained on the Syn-Train dataset and evaluated zero-shot on the unseen large roadnet.
>
> **Table 2. Zero-shot performance comparison on Large Roadnet (ATT / AWT / AQL, lower is better)**
>
> | Model            | ATT     | AWT     | AQL       |
> |------------------|---------|---------|-----------|
> | FixedTime        | 1901.82 | 885.05  | 14891.94  |
> | MaxPressure      | 1165.76 | 119.38  | 9223.00   |
> | MPLight          | 1392.39 | 189.82  | 12013.66  |
> | AdvancedCoLight  | 1195.48 | 333.31  | 9395.88   |
> | CoLight          | 1380.24 | 207.46  | 11872.34  |
> | LLMLight-8B      | 1111.76 | 144.98  | 8257.82   |
> | **CoLLMLight-8B**| **1008.10** | **86.28** | **6902.18** |
>
> CoLLMLight-8B achieves the best performance across all three metrics on this large-scale scenario. The improvement margin is consistently larger than conventional MARL baselines, indicating that the semantic reasoning mechanism maintains robustness and generalization capability even under highly complex urban networks.
>
> # Closing note
> We thank the reviewer once again for their valuable feedback, which has greatly helped us improve the quality and clarity of the work.
> We hope that the provided clarifications, analyses, and new experimental results adequately address all concerns and will be viewed favorably in the final evaluation.
> We would be happy to provide further explanations if needed.

---

### Author Response · Authors · 2025-12-02
**Official Comment to Program Chair and Area Chair Part 2**

**Reviewer TspF (Rating: 4)**

- **Novelty and contributions of CoLLMLight:** The reviewer raised concerns regarding the novelty and contributions of our work. We clarified that CoLLMLight introduces three fundamental contributions: achieving network-wide optimization via **the first cooperative LLM framework for TSC**, ensuring real-time feasibility through an Asynchronous Cooperative Decision Architecture, and balancing efficiency via Cost-Aware Cooperation Optimization.

- **Ablation Study & Cooperation Dynamics:** The reviewer questioned what the LLM learns from the spatiotemporal context and how its reasoning benefits cooperation. We provided an ablation study on the spatiotemporal context and illustrated the cooperation process with a concrete example. We added **Appendix A.7 Case Study** and **Appendix A.11 Spatiotemporal Ablation Study**.

- **Real-time Feasibility and Deployment:** The reviewer expressed concerns regarding real-time feasibility and deployment considerations. We analyzed communication latency, bandwidth, and inference time to demonstrate the practical feasibility of our method. We added a detailed discussion in **Appendix A.14 Feasibility of Real-World Deployment**.

- **Variance and Statistical Significance Analysis:** The reviewer requested an analysis of variance and statistical significance. We provided detailed experimental results including statistical metrics. We added **Appendix A.8 Statistical Significance Analysis**.

- **New Baselines:** The reviewer suggested comparing with more recent baselines. We incorporated three new baselines, including X-Light, a multi-agent transformer-based method. We updated the experimental results to reflect these comparisons.

- **Human Evaluation:** The reviewer suggested including a human evaluation. We conducted an expert inspection study involving four transportation researchers familiar with traffic signal control. We included the findings in **Appendix A.12 Human Evaluation**.

- **Clarification of SR Reasoning and Communication:** The reviewer asked for clarification on the format of SR reasoning traces and communication content. We clarified that SR reasoning traces are used in natural language form, while the communication mechanism exchanges numerical traffic state data.

- **Robustness Evaluation:** The reviewer asked about the robustness of CoLLMLight under stale SR and communication failure scenarios. We conducted specific experiments to demonstrate the system's robustness under these conditions. We added **Section 4.6 Robustness Evaluation** to the main text.

---

**In summary**, **Reviewers jThr, Vb9x, and 9P3T have shown explicit support**, highlighted by an 8 from Reviewer 9P3T and a score increase (4$\rightarrow$6) from Reviewer Vb9x. Regarding Reviewer TspF, we have conducted substantial new experiments and analyses that we believe fully address their concerns. We respectfully request that the AC/PC consider this positive review trajectory and the comprehensive revisions made.

Thank you for your consideration.

Best regards,

The Authors of Submission 7015

---

### Author Response · Authors · 2025-12-02
**Official Comment to Program Chair and Area Chair Part 1**

**Dear Program Chair and Area Chair,**

We sincerely appreciate your oversight throughout the review process and the reviewers’ constructive feedback, which has improved the clarity, rigor, and presentation of our work. Below we summarize the reviewers’ concerns, our responses, and the corresponding revisions. We believe all concerns have been fully addressed in either the original submission or the revised main text.

---
**Reviewer jThr (Rating: 6)**

- **Methodological details:** The reviewer asked how Cooperative Reasoning and Reflection are implemented in CoLLMLight. We clarified their roles and provided a detailed CoLLMLight procedure. We revised Section 3.1.2 (Spatiotemporal-aware Cooperative Reasoning) to provide clearer detailed descriptions. We added **Appendix A.10 CoLLMLight Pseudocode**.

- **Neighborhood radius, lanes, and deployment considerations:** We clarified the precise definition of neighborhood radius and lane sets, and provided analyses on bandwidth, latency, and cost. We added **Appendix A.9 Communication Mechanism** and **Appendix A.14 Feasibility of Real-World Deployment**.

- **Larger-network test:** The reviewer requested testing on larger networks. We discussed how CoLLMLight scales with road network size and added results on a large Manhattan network. We added **Appendix A.13 Experiment on Large Road Networks**.

---
**Reviewer Vb9x (Rating: 4 -> Increase to 6)**

- **Methodological details:** The reviewer inquired about the base model specifications and requested clarification of the CoLLMLight workflow. We clarified that the base model is Llama-3.1-8B and provided the detailed CoLLMLight procedure. We specified the base model in the main text and added **Appendix A.7 Case Study** and **Appendix A.10 CoLLMLight Pseudocode**.

- **Supervised Performance Comparison:** The reviewer requested a comparison with RL-based methods in a setting where training and evaluation occur on the same road network. We explained our experimental setting and highlighted the results on the Syn-Train dataset (**Appendix A.6** of the original submission), demonstrating the competitive supervised performance of CoLLMLight.

- **Framework Effectiveness:** The reviewer questioned the overall effectiveness of our framework. To address this concern, we provided a controlled comparison showing CoLLMLight significantly outperforms the LLMLight base model (Llama-3.1-8B) both before and after optimization. We added the results of LLMLight (Llama-3.1-8B) to **Table 1** in the main text.

**Follow-up:** The reviewer explicitly stated that the new experiments were helpful and **confirmed they would increase their score.**

---
**Reviewer 9P3T (Rating: 8)**

- **Discussion of LLM and MARL-based methods:** The reviewer asked for a justification of using LLMs over conventional MARL methods. We clarified that the semantic reasoning capability of LLMs avoids the overfitting common in the implicit cooperation learned by conventional MARL, enabling superior zero-shot transfer. We added relevant discussions to the **Experiments** and **Related Work** sections.

- **New Baselines:** The reviewer requested comparisons with more recent baselines. We conducted experiments including three recent MARL baselines, where CoLLMLight achieved the best performance across all metrics. We included these new comparative results in the revised manuscript.

- **Asynchronous Consistency:** The reviewer raised concerns about whether SR latency causes decision desynchronization. We explained that the SR-RD decision pipeline ensures decision consistency and provided robust tests under stale SR and communication failure scenarios. We added **Section 4.6 Robustness Evaluation** to the main text.

- **Explicit Cooperation Mechanism:** The reviewer suggested a more explicit cooperation mechanism. We explained that our cooperation policy is already explicit and interpretable, and we discussed the potential for further explicit mechanisms. We added **Appendix A.7 Case Study** to demonstrate this interpretability.

---

### Meta-Review · Area_Chair_wx9Y · 2026-01-04

**Summary:**

This paper proposes CoLLMLight, a cooperative LLM-based framework for network-wide traffic signal control that extends prior single-intersection LLM agents to a multi-agent setting. The key ideas are an asynchronous reasoning–decision architecture that decouples heavy spatiotemporal reasoning from real-time control, and a cost-aware cooperation optimization that balances coordination quality with token and latency constraints. Experiments on multiple real-world traffic networks validate consistent improvements over rule-based, MARL-based, and prior LLM-based baselines, with some level of zero-shot generalization.

Reviewers questioned whether the method represents a fundamentally new learning contribution or a well-engineered extension of prior work (especially LLMLight), noting that several components feel incremental, and AC agrees. There were also concerns about clarity and rigor, particularly around how cooperative reasoning actually works, how asynchronous decisions remain consistent across intersections, and whether the gains come from the cooperative framework itself or mainly from policy refinement and training. Finally, some reviewers were initially skeptical about real-time feasibility, generalization beyond simulation, and the lack of deeper analysis explaining why LLM-based semantic reasoning leads to better coordination, though many of these issues were partially addressed in the rebuttal with lots of added experiments, numbers and clarifications.

The contribution is high in the traffic control with LLM domain, so I would suggest accept with poster but would be okay seeing it as an oral paper.

**Reviewer Concerns:**

The first is novelty judgement. Even though the authors clarified the framework and added strong experimental evidence, the core ideas still feel like a careful extension of earlier LLM-based traffic control work rather than a fundamentally new learning approach. Cooperation, asynchronous reasoning, and cost-aware control are well designed and practically useful, but they come across more as system and engineering advances than a conceptual shift in how multi-agent traffic control is learned. Meanwhile, with the progress of the field, AC thinks system and engineering advances should be recognized in ICLR as well.

The rebuttal added ablations and concrete case studies showing that spatial and temporal context matters, which helps, but the explanation of why LLM-based semantic reasoning leads to better coordination than strong graph-based methods remains mostly just intuition. The paper shows performance gains, especially in zero-shot settings, but it still offers limited insight into the underlying coordination dynamics or when this approach would fail. Final draft AC would encourage the authors to pay attention to beefing up these analysis.

**Reviewer Scores:**

Vb9x raised to 6

TspF is likely to raise as well, to 6 maybe.

So final: 6 6 6 8

---

### Decision · Program_Chairs · 2026-01-26

Accept (Poster)